# Efficient coding of natural images in the mouse visual cortex

Federico Bolaños[1], Javier G. Orlandi [2] ✉, Ryo Aoki[3], Akshay V. Jagadeesh[4], Justin L. Gardner [4] & Andrea Benucci [3,5] ✉

How the activity of neurons gives rise to natural vision remains a matter of intense investigation. The mid-level visual areas along the ventral stream are selective to a common class of natural images—textures—but a circuit-level understanding of this selectivity and its link to perception remains unclear. We addressed these questions in mice, first showing that they can perceptually discriminate between textures and statistically simpler spectrally matched stimuli, and between texture types. Then, at the neural level, we found that the secondary visual area (LM) exhibited a higher degree of selectivity for textures compared to the primary visual area (V1). Furthermore, textures were represented in distinct neural activity subspaces whose relative distances were found to correlate with the statistical similarity of the images and the mice's ability to discriminate between them. Notably, these dependencies were more pronounced in LM, where the texture-related subspaces were smaller than in V1, resulting in superior stimulus decoding capabilities. Together, our results demonstrate texture vision in mice, finding a linking framework between stimulus statistics, neural representations, and perceptual sensitivity—a distinct hallmark of efficient coding computations.

Visual textures are broadly defined as "pictorial representations of spatial correlations"[1]—images of materials with orderly structures and characteristic statistical dependencies. They are pervasive in natural environments, playing a fundamental role in the perceptual segmentation of the visual scene[1,2]. For example, textures can emphasize boundaries, curvatures[3,4], 3D tilts, slants[5,6] and distortions, support a rapid "pop-out" of stimulus features[7], and can form a basis set of visual features necessary for object vision[8].

Although texture images largely share the spectral complexity of other natural images[9-11], they can be more conveniently parametrized and synthetized than other natural images. This has been explored via diverse computational approaches: in the field of computer graphics[12], via entropy-based methods[13-15], using wavelet approaches[16,17], and, more recently, in machine learning implementations based on deep convolutional neural networks[18-21].

In light of their rich statistics and convenient synthesis and parametrization, texture images have been at the core of studies on efficient coding principles of neural processing. According to one interpretation of the efficient coding hypothesis[22], the processing of visual signals along hierarchically organized cortical visual areas reflects the statistical characteristics of the visual inputs that these neural circuits have learned to encode, both developmentally and evolutionarily[23-29]. Accordingly, texture images have been extensively used in experimental studies that have examined the contribution of different visual areas to the processing of texture statistics.

In particular, studies in primates have revealed that the "mid-level" ventral areas, V2–V4, are crucial for processing texture images[30-41], more so than the primary visual cortex, V1 (however, see ref. 42). Furthermore, as revealed by psychophysical observations[43] and neural measurements, area V2, in addition to being differentially

[1]University of British Columbia, Neuroimaging and NeuroComputation Centre, Vancouver, BC V6T, Canada. [2]University of Calgary, Department of Physics and Astronomy, Calgary, AB T2N 1N4, Canada. [3]RIKEN Center for Brain Science, Laboratory for Neural Circuits and Behavior, Wakoshi, Japan. [4]Stanford University, Wu Tsai Neurosciences Institute, Stanford, CA, USA. [5]Queen Mary, University of London, School of Biological and Behavioral Science, London E1 4NS, UK. ✉e-mail: javier.orlandi@ucalgary.ca; andrea@benuccilab.net

modulated by the statistical dependencies of textures, correlates with the perceptual sensitivity for these stimuli[34,35,38]. Notably, biology-inspired computational studies using artificial neural networks have similarly emphasized hierarchical coding principles, with V2-like layers as the locus for representing texture images in classification tasks[44,45]. Together, these observations suggest a general hierarchical coding framework, where the extrastriate visual areas, in particular area V2, define a neural substrate for representing texture stimuli, reflecting a progressive elaboration of visual information from "lower" to "higher" areas along the ventral visual stream.

This high-level view raises two fundamental questions: (1) whether this coding framework applies, in all generality, to hierarchically organized visual architectures as seen in several mammalian species other than primates—as CNN simulations would suggest—and (2) which functional principles at the circuit level give rise to texture selectivity, especially in the secondary visual area V2. Both questions hinge on the need to gain a computational and mechanistic understanding of how visual networks process naturalistic statistical dependencies to enable the perception of scenes and objects[1,2,46–48].

Addressing these questions in the mouse model organism would be particularly advantageous[49]. Although the rodent visual system is much simpler than that of primates[50], mice and rats have a large secondary visual cortex (area LM) homologous to primate V2[51,52], belonging to a set of lateral visual areas forming a ventral stream of visual processing[53,54]. As recordings from these areas have revealed, there is increased selectivity for complex stimulus statistics in both rats[55,56] and mice[57,58]. Therefore, we studied the processing of texture images in mice with an emphasis on the interrelationship between behavioral, neural, and stimulus-statistic representations. Using a CNN-based algorithm for texture synthesis[59], we generated an arbitrary number of naturalistic texture exemplars and "scrambles"—spectrally matched images lacking the higher-order statistical complexity of textures[48,60–63]—by precisely controlling the statistical properties of all the images. Using these images, we demonstrated that mice can perceptually detect higher-order statistical dependences in textures, distinguishing them from scrambles, and discriminating among the different types of naturalistic textures ("families" hereafter). At the neural level, using mesoscopic and two-photon GCaMP imaging, we found that the area LM was differentially modulated by texture statistics, more so than V1 and other higher visual areas (HVAs). Examining the representational geometry of the population responses, we found that when the statistical properties of a texture were most similar to those of scrambles, the corresponding neural activity was also more difficult to decode, and the animal's performance decreased. These dependencies were particularly prominent in LM and when considering the higher-order statistical properties of the images. Notably, LM encoded different texture families in neural subspaces that were more compact than in V1, thus enabling better stimulus decoding in this area.

## Results

### Training mice to detect and discriminate between texture statistics

To examine the ability of mice to use visual–texture information during perceptual behaviors, we designed two go/no go tasks. In the first task, mice had to detect the texture images interleaved with scramble stimuli. In the second task, mice had to discriminate between two types of texture images from different texture families.

**Synthesis of textures and scrambles.** We generated synthetic textures using an iterative model that uses a convolutional neural network (VGG16) to extract a compact multi-scale representation of texture images[59] (Fig. 1a). To disentangle the contribution of higher-order image statistics from lower-order ones, for each texture exemplar we synthesized a spectrally matched image (scramble, Fig. 1b) having the same mean luminance, contrast, average spatial frequency, and orientation content (Supplementary Fig. 1a–c, Methods) but lacking the higher-order statistical features characteristic of texture images. This produced image pairs for which the main axis of variation was higher-order statistics (textural information). In total, we synthesized images belonging to four texture families and four associated scramble families, each with 20 exemplars.

**Behavioral detection of texture statistics.** To train the mice in the two go/no go tasks, we employed an automated training setup[64,65], wherein the mice were asked to self-head fix and respond to the visual stimuli displayed on a computer screen located in front of them (Fig. 1c). Mice were trained to respond to the target stimuli by rotating a toy wheel, and contingent on a correct response, they were rewarded with water. For the texture/scramble go/no go task, the "go" stimuli were texture images, while the "no go" stimuli were image scrambles (Fig. 1d). For responses to a no-go stimulus (false alarms), a checkerboard pattern was displayed on the screen for 10 s before a new trial began. All mice (n = 19) learned the task in approximately 25 days (i.e., the time needed for d-prime > 1 in at 50% of the mice, Fig. 1e-g). Mice could significantly discriminate between all four texture/scramble pairs (Fig. 1f, d' > 1, $p < 0.05$ for all families, one sample t-test with Holm-Bonferroni correction for multiple comparisons; Supplementary Table 1) with an average discriminability value of d' = 2.1 ± 0.15 (s.e.), and with the "rocks" family having a significantly lower performance than all other families, both within and across mice. Dissecting the animals' performance, we found that, on average, mice had a high proportion of hits (Supplementary Fig. 2a), as expected given that the training procedure encouraged "go" behaviors[66], with the lowest performance for rocks associated with a higher proportion of false alarms (Supplementary Fig. 2b). Additionally, to ensure that the mice were not adopting a strategy based on "brute force" memorization (e.g., of pixel-level luminance features[67]), we synthesized a novel image set consisting of 20 new exemplars for each of the four families, together with corresponding scramble images. Then, in a subset of the mice (n = 4 for scales; n = 3, rocks; n = 11, honeycomb; n = 8, plants), we switched the original set of images with the novel set and compared behavioral performance between the last five sessions prior to the switch and the five sessions after the switch, finding no significant difference (Supplementary Fig. 2c).

**Behavioral discrimination between texture families.** Mice not only could detect higher-order statistical features in texture images that were missing in the scrambles, but they could also discriminate between different texture statistics. We trained mice already expert in texture–scramble discrimination, as well as a new cohort of naïve mice (n = 2), in a second go/no go task (Supplementary Table 2). Mice were shown exemplars (n = 20) from two texture families, randomly chosen but fixed across sessions, with only one of the two families associated with a water reward for a correct "go" response. In addition, all 40 exemplars were randomly rotated to prevent mice from solving this task using orientation information that may have been different across families (Supplementary Fig. 2d). Mice could discriminate between texture families, with a significantly positive d' for all six texture pairs (Fig. 1h, d' > 0, $p < 0.019$ for all pairs, one sample t-test with Holm-Bonferroni correction at α = 0.05).

Finally, we controlled that in both tasks mice were not relying on "simple" statistics, such as the skewness and kurtosis of the luminance histogram, with skewness having been previously related to texture perception (e.g., the *blackshot* mechanism[61]). For this, we created a new set of textures and scrambles in which skewness and kurtosis values were randomly mixed between textures and scrambles, thus uninformative of the texture family and texture-scramble identity (Supplementary Fig. 3a), finding that behavioral performance was unaffected by this manipulation (Supplementary Fig. 3b). This result

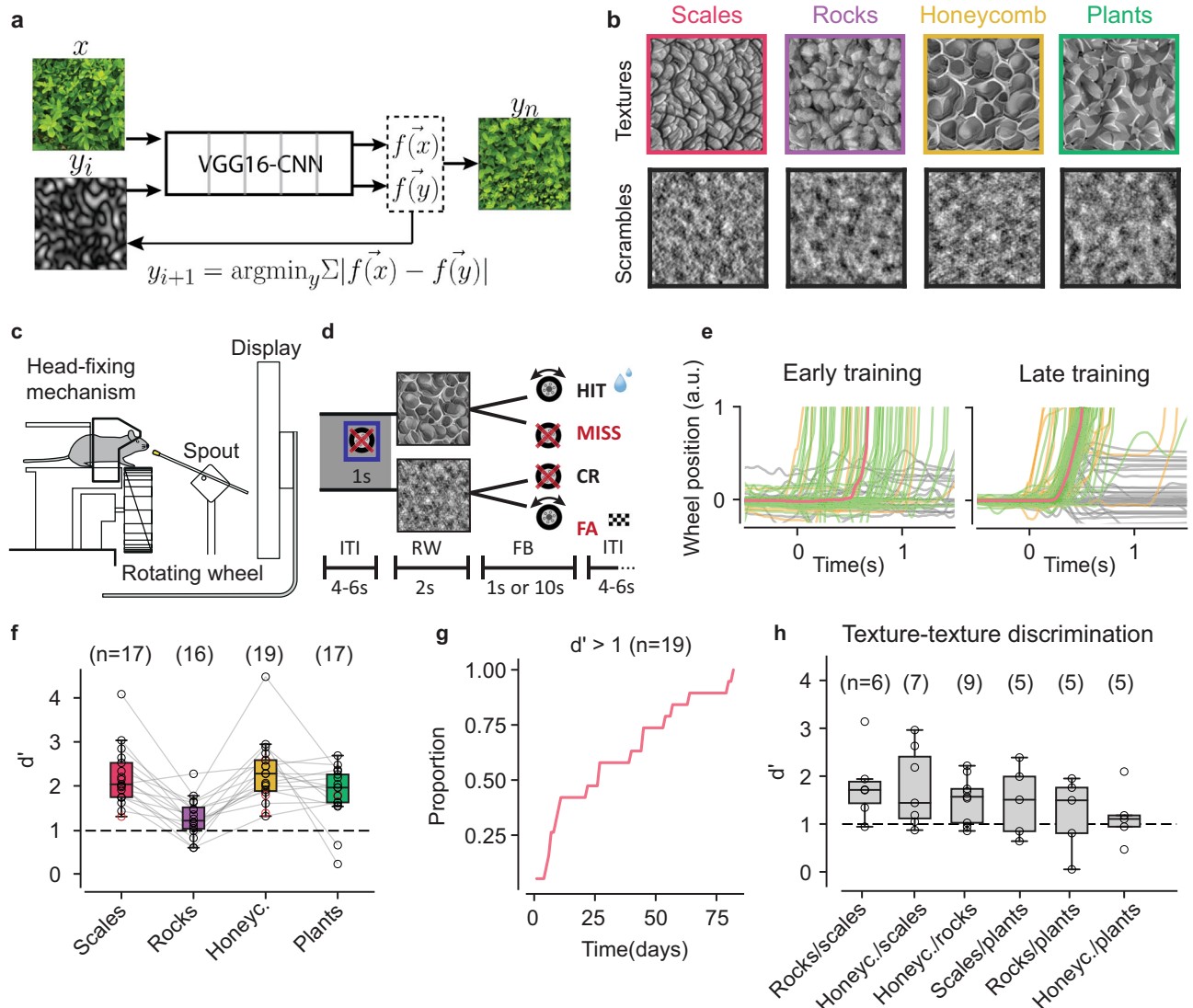

**Fig. 1 | Mice can discriminate texture statistics from spectrally matched scrambles and between texture families. a** Schematic plot of the iterative algorithm to synthetize the texture images based on the VGG16-CNN architecture; 'x', target texture; 'f(x)', texture representation by the network; 'y' is an initial 'seeding' Gaussian-noise image with 'f(y)' being its network representation. The optimization minimizes the difference between f(x) and f(y) by iteratively changing 'y_i' to obtain 'y_n'. **b** Examples of texture families and the respective spectrally matched stimuli (scrambles). **c** Schematic plot of the automatic training system with self-head fixation. **d** Texture – scramble go/no go task: the mouse must rotate a rubber wheel (go trial) if shown a texture exemplar; it must keep it still if shown a scramble (no go). ITI is the inter-trial interval; RW, the response window; and FB, the feedback period. **e** Representative examples of wheel rotations from an early training session (left) and a well-trained mouse (right); green for hits, yellow for false alarms, gray for either misses or correct rejects, and orange for the average across hits. **f** Behavioral discriminability (d') in the texture–scramble task for expert mice for

each family; colors as in **b**. The top labels are the number of mice trained in each of the families; $n = 16$ out of 19 mice were trained in all the family-scramble pairs (connecting gray lines). The empty dots indicate the individual animals. Rocks have the lowest performance: one-way ANOVA across families, $p = 2 \times 10^{-5}$; post-hoc Tukey HSD, rocks vs. honeycomb: $p = 3 \times 10^{-5}$, rocks vs. plants: $p = 0.042$, rocks vs. scales: $p = 0.0002$, $n = 16$ mice. Rocks, d' = $1.4 \pm 0.14$ tested for d' > 1 (horizontal broken line), $p = 0.016$, one sample t-test, two-sided. **g** The time needed for mice (proportion of days) to reach d' > 1 in their first family-scramble training. **h** Behavioral discriminability (d') in the texture–texture task across all six possible pairs of the four families. The top labels are the number of mice trained in each texture pair; broken horizontal line as in (f). Each animal was trained in a different number of family pairs (Supplementary Table 1, 2). Box plots in (**f**, **h**) and in other figures indicate the median with a horizontal bar; the box height denotes the inter-quartile range (IQR, 1st and 3rd quartile) and the whiskers extend by 1.5 x IQR. Source data are provided as a Source Data file.

shows that the heuristic[68] employed by the mice was not based on these simple statistics, supporting the interpretation that mice relied on the high-order spatial correlation properties of texture images.

## Widefield responses to textures and scrambles
To examine the neural activity underlying the mice's ability to detect and discriminate between texture statistics, we imaged multi-area responses from the posterior cortex of untrained animals whose neural dynamics were unaffected by procedural or perceptual learning processes. This choice assumes that texture processing in visual

cortical networks is likely not the outcome of our behavioral training (see also Discussion).

We performed widefield calcium imaging during the passive viewing of textures and scrambles. Mice ($n = 11$) were placed in front of a computer screen that displayed either an exemplar of a texture or a scramble (Fig. 2a). The stimuli, 100 degrees in size, were presented in front of the mice, centered on the mouse's body midline, as was done for behavioral training. While mice passively viewed the stimuli, we recorded both calcium-dependent and calcium-independent GCaMP responses using a dual wavelength imaging setup. We then used the

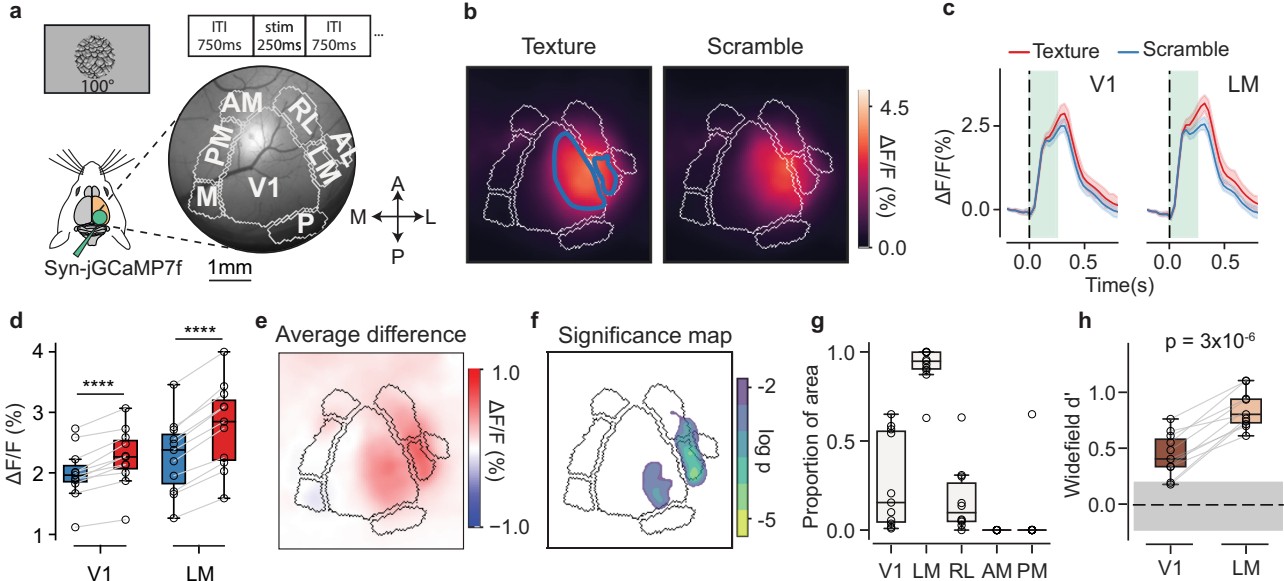

**Fig. 2 | Texture stimuli differentially modulate V1 and LM at the mesoscale level. a** Schematic plot of the widefield imaging setup. Top, an example of a texture image (100° diameter) and the stimulus presentation times. Right, a representative example of the right posterior cortex of a mouse with main-area borders (gray lines). V1, primary visual cortex; LM, secondary visual cortex (lateromedial); RL, rostrolateral; AL, anterolateral; AM, anteromedial; PM, posteromedial; M, medial area; P, posterior area. **b** Average GCaMP response (across all exemplars and repeats) to texture stimuli (left) and scramble stimuli (right) for a representative example mouse. The blue contours show the regions of interest (ROIs) retinotopically matching the visual stimuli in V1 and LM. **c** Average GCaMP responses from the retinotopically matched ROIs in V1 and LM for the same representative example shown in (b); texture–scramble difference, %ΔF/F, V1: 0.27% ± 0.04%; LM: 0.50% ± 0.03%. Textures in red, scrambles in blue. The green-shaded rectangles show the stimulus duration (250 ms); the vertical broken line indicates the time of stimulus onset. Shaded bands are 95% CIs (4 families x 20 exemplars). **d**, Responses across mice to textures and scrambles in V1 and LM ROIs; V1: scramble, ΔF/F: 2.0% ± 0.1% s.e., texture: 2.2% ± 0.1%; $p = 3 \times 10^{-5}$, two-sided

paired t-test, n = 11 mice. For LM: scramble: 2.3 ± 0.2, texture: 2.8 ± 0.2; p = 1 × 10⁻⁶. Color code as in (c). Box plots indicate the median with a horizontal bar; the box height denotes the inter-quartile range (IQR, 1st and 3rd quartile) and the whiskers extend by 1.5 x IQR. **e**, The difference between the texture–scramble images shown in (b). **f**, The regions with a statistically significant response difference shown in (e) – the logarithm of the p-values from a two-sided t-test; colored regions for p < 0.01. **g**, Proportion of pixels in each visual area (within the retinotopically identified ROIs) modulated by the textures relative to the scrambles (n = 11 mice, empty dots); V1: 0.27 ± 0.08 s.e., LM: 0.92 ± 0.03; RL: 0.17 ± 0.06; AM: not detected; PM: 0.06 ± 0.06. Box plots as in (d). **h** Discriminability measure (d′) between textures and scrambles from ΔF/F (%) responses within the same ROIs used for (g). The gray horizontal band corresponds to a null d′ distribution derived from pre-stimulus activity. Each dot is one animal, connecting lines for the same-mouse data. The horizontal broken line indicates the mean of the null distribution; p-value from two-sided paired t-test. For statistical significance in this and other figures, * is for $p < 10^{-2}$, ** $p < 10^{-3}$, *** $p < 10^{-4}$. **** $p < 10^{-5}$. Box plots as in **d**. Source data are provided as a Source Data file.

calcium-independent GCaMP response to correct for the hemodynamic component of the calcium-dependent GCaMP responses[69]. We recorded from the right posterior cortex, which gave us access to ~5–6 HVAs (Fig. 2a). All the reliably segmented HVAs retinotopically represented the stimulus position in visual space (Supplementary Fig. 1d).

The peak-response maps to the textures and scrambles showed activations almost exclusively in V1 and LM (Fig. 2b). When averaging within the ROIs retinotopically matching the visual stimuli (example blue contours in Fig. 2b), the responses were larger for textures than scrambles both in V1 and LM (Fig. 2c, d) and accordingly, the difference in the peak-response maps resulted in a differential modulation localized primarily in V1 and LM (Fig. 2e). To establish statistical significance, we tested the modulation of each pixel against a null distribution derived from the pre-stimulus period (Fig. 2f), and to determine the significance of an entire visual area, we computed the proportion of significantly modulated pixels in each area within retinotopic ROIs, demonstrating that areas V1 and LM were those most prominently modulated by textures relative to scrambles (Fig. 2g). To compare response modulations between V1 and LM, we computed a texture discriminability measure (d′) in retinotopically matched ROIs and found that d′ values in LM were significantly higher than those in V1 (Fig. 2h. V1: 0.41 ± 0.05, s.e.; LM: 0.79 ± 0.05; difference, $p = 3 \times 10^{-6}$, paired t test, n = 11 mice).

Finally, we observed that, despite the stimuli being retinotopically mapped onto the central-lateral portion of V1 (Fig. 2b, blue contours), significant texture-scramble modulations were most prominent in the

posterior-lateral region of V1 (example in Fig. 2f). To test for a possible representational gradient or asymmetry in spatial representations[70–72], we performed experiments with full-field texture and scramble images presented monocularly on a monitor sufficiently large as to activate the entire V1 (azimuth, [−62.4°, +62.4°]; elevation [−48.5°, +48.5°]). Based on maps of elevation and azimuth, we then divided V1 into four quadrants representing the left-right upper and lower visual fields, finding that texture discriminability (d′) was consistently higher in the upper visual field (Supplementary Fig. 4).

Together, these results indicate that, at the mesoscopic level, texture selectivity in V1 is biased toward the upper visual field, and when considering a constellation of HVAs surrounding the primary visual cortex, LM is the area with the most significant selectivity to higher-order texture statistics.

### Single-cell responses to texture and scrambles

**Proportion of cells responding to textures in V1 and LM.** We examined the circuit-level representations underlying this mesoscale selectivity using two-photon GCaMP recordings in areas V1 and LM (Fig. 3a). Imaging ROIs (approximately 530 μm x 530 μm) in V1 and LM were selected based on the retinotopic coordinates of the visual stimuli, and neural activity was recorded while presenting three classes of visual stimuli: static gratings of different orientations and spatial frequencies (sf; four orientations spaced every 45 degrees, 100° in size, full contrast, sf = [0.02, 0.04, 0.1, 0.2, 0.5] cpd), scrambles and texture images matching the properties of the stimuli used in behavioral

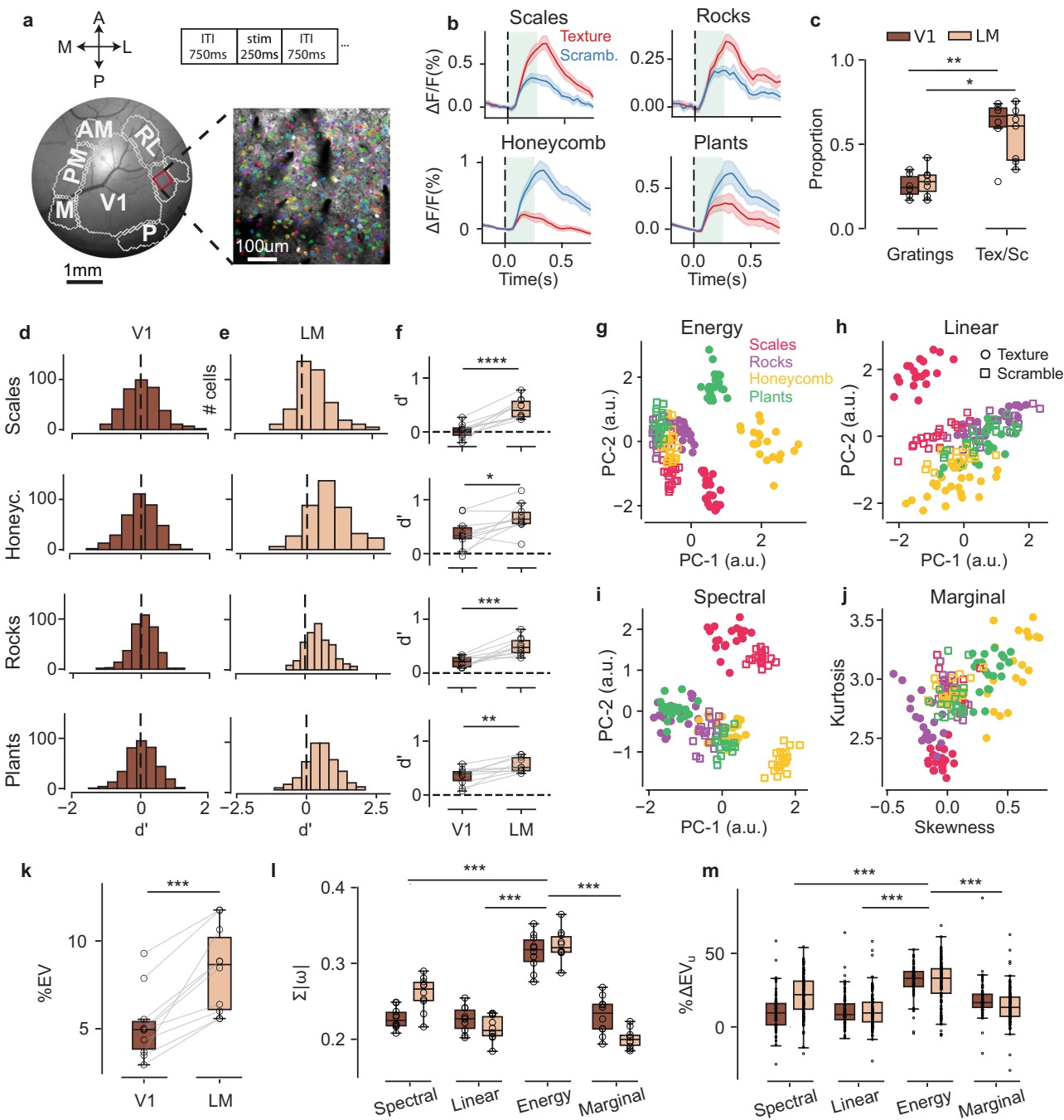

experiments (four families for scrambles and textures, each with 20 exemplars rotated either by 0 or 90 degrees, and with eight repetitions of each image).

The single-cell responses to oriented gratings agreed with what is typically reported in the literature (e.g. refs. 73,74), with approximately 25–30% of the segmented cells being visually responsive (Fig. 3c). The responses to textures and scrambles were rather heterogenous, with some cells strongly responding to textures, others to scrambles, and several showing mixed selectivity (Fig. 3b). In both V1 and LM, there was a significantly larger proportion of cells responding to textures or scrambles relative to gratings (Fig. 3c, V1, textures or scrambles: 61% ± 6%; LM: 55% ± 6%, s.e.). Despite the significant heterogeneity, responses averaged across cells were significantly larger in LM than in V1 for all texture families (Supplementary Fig. 5a, b). We then quantified the

texture–scramble response modulation of the individual cells using a discriminability measure (d′), similar to what was done in mesoscale analyses (Fig. 3d, e), and found that (i) the proportion of cells with significantly positive d′ values (i.e., with larger values in response to textures) were higher in LM than in V1 for all families (Supplementary Fig. 6c); (ii) the average d′ value was higher in LM than V1 for all families (Fig. 3f, V1: average d′ = 0.24 ± 0.01, LM: average d′ = 0.54 ± 0.01, p = 2 × 10⁻⁴, paired t-test, n = 10; Supplementary Fig. 6a,b), which reflected larger response amplitudes to textures than scrambles (Supplementary Fig. 6d).

Together, these results indicate that underlying the increased widefield texture selectivity in LM is both an increase in the proportion of texture-selective cells as well a larger texture–scramble modulation of individual cells.

**Fig. 3 | Single-cell responses in LM better discriminate textures from scrambles.** **a** Multi-area imaging, as in Fig. 2a, with inset showing a representative ROI for two-photon recordings; colored dots indicate the segmented cells responsive to textures and/or scrambles ("stim", top). **b** Top panels: two example cells responding more strongly to a texture family (red) than scrambles (blue); bottom panels, two example cells for the opposite selectivity. Shaded bands are 95% CIs; center is the mean across 80 texture samples (20 samples per family); vertical broken line indicates the time of stimulus onset, and the green rectangles show the stimulus duration (250 ms). **c** The proportion of cells that significantly responded to oriented gratings in V1 and LM (V1: 25% ± 3% s.e., $n = 6$ mice; LM: 28% ± 3%, n = 7; average no. of segmented cells = 381 ± 44 in V1 and 344 ± 46 in LM) was lower that the proportion of cells responding to either textures or scrambles (Tex/Sc; V1: 61% ± 6% s.e., LM: 55% ± 6%). Gratings vs. Tex/Sc in V1: $p = 0.002$, LM: $p = 0.015$, two-sided paired t-test. Box plots indicate the median with a horizontal bar; the box height denotes the inter-quartile range (IQR, 1st and 3rd quartile) and the whiskers extend by 1.5 x IQR. Distributions of the texture–scramble discriminability values (d′) computed for each cell. Rows are for texture families; **d** for V1, **e** for LM: colors as in **c**. Data from a representative experiment. **f** The mean d′ values for all the experiments in V1 and LM ($n = 10$ mice, open dots); connecting lines for the same-mouse data; V1 d′: scales = 0.02 ± 0.04 s.e., rocks = 0.21 ± 0.03, honeycomb = 0.37 ± 0.08, plants = 0.35 ± 0.05, LM d′: scales = 0.43 ± 0.05, rocks = 0.50 ± 0.05, honeycomb = 0.67 ± 0.08, plants = 0.56 ± 0.04; p-values from two-sided paired t-tests with Holm-Bonferroni correction: scales, $p = 7 \times 10^{-5}$; rocks, $p = 6 \times 10^{-4}$; honeycomb, $p = 2 \times 10^{-2}$; plants, $p = 4 \times 10^{-3}$ ($n = 10$). Box plots as in **c**. **g–j** Two-dimensional PCA embedding of each of the four groups of image statistics (titles). The dots indicate texture exemplars (20), and the squares scramble exemplars (20). Color code for

texture families in the legend. The same images were used for both behavioral and imaging experiments. **k** The explained variance (EV, %) by the encoding linear model based on PS image statistics, comparing V1 to LM; only cells for which EV ≥ 1% have been included in the analysis (permutation test, Methods); each dot is a mouse; connecting lines for the same-mouse data; V1: 5.3% ± 0.6% s.e., LM: 8.2% ± 0.7%, cross-validated, $n = 10$, V1 – LM difference, $p = 7 \times 10^{-4}$, two-sided paired t-test. Box plots as in **c**. **l** The sum of weight values for each of the PS statistic groups of the fitted regressive model; each dot is an average across cells for a given mouse. The energy statistics are significantly higher than all others, both in V1 and in LM: $p < 0.001$ for energy compared to all other statistics V1 comparisons: energy vs. linear: $5 \times 10^{-8}$; energy vs. marginal: $1 \times 10^{-6}$; energy vs. spectral: $1 \times 10^{-6}$; linear vs. marginal: 0.7; linear vs. spectral: 0.7; marginal vs. spectral: 0.9. LM comparisons: energy vs. linear: $1 \times 10^{-12}$; energy vs. marginal: $2 \times 10^{-13}$; energy vs. spectral: $3 \times 10^{-6}$; linear vs. marginal: 0.8; linear vs. spectral: $5 \times 10^{-5}$; marginal vs. spectral: $4 \times 10^{-6}$.: one-way ANOVA with post-hoc analysis (Tukey HSD). Colors and box plots as in **c**. **m** The "unique" EV (%ΔEV$_u$) for all four PS statistics groups. The cells with a high explained variance by the full model (EV ≥ 10%) were included in the analysis. Each dot is the change in explained variance for a cell when using the "full" model or a model missing a given PS statistic. The energy statistics are significantly higher than all others, both in V1 and LM: $p < 0.001$ for energy larger than all other statistics; V1 comparisons: energy vs. linear: $<10^{-12}$; energy vs. marginal: $7 \times 10^{-12}$; energy vs. spectral: $<10^{-12}$; linear vs. marginal: $1 \times 10^{-4}$; linear vs. spectral: 0.9; marginal vs. spectral: $1 \times 10^{-4}$. LM comparisons: energy vs. linear: $5 \times 10^{-15}$; energy vs. marginal: $5 \times 10^{-15}$; energy vs. spectral: $8 \times 10^{-8}$; linear vs. marginal: 0.04; linear vs. spectral: $3 \times 10^{-11}$; marginal vs. spectral: $6 \times 10^{-5}$; one-way ANOVA with post-hoc analysis (Tukey HSD). Box plots as in **c**. Source data are provided as a Source Data file.

**Encoding linear model of neural responses.** To isolate the set of statistical features that most prominently drove the texture–scramble selectivity in V1 and LM, we used a previously described mathematical model to parametrize image statistics: the Portilla–Simoncelli statistical model (henceforth, PS model and statistics[15]). This model employs a set of analytical equations to compute the correlations across a set of filters tuned to different image scales and orientations. These statistics can be divided into four main groups: marginal (skewness and kurtosis of the pixel histogram), spectral, linear cross-correlation, and energy cross-correlation statistics. In its complete formulation, the model provides a very high dimensional parametrization of the stimuli (740 parameters), resulting in more parameters than the total number of images (320). Therefore, dimensionality reduction of each PS group of statistics can provide a reduced representation without significant loss in parametrization power (Supplementary Fig. 8, 9). We used Principal Component Analysis (PCA), finding that with even the first two principal components, the energy statistics are best at separating textures from scrambles and between texture types (Fig. 3g–j, Supplementary Fig. 7, 9), as also reported in human psychophysical studies[34].

Using PS statistics as features, we created an encoding linear model for single-cell responses in V1 and LM. The model's task was to predict the response of a particular neuron to all texture and scramble exemplars as a weighted linear sum of PS coefficients. When considering the cells for which the model could explain at least 1% of the response variance—that is, a threshold value for the significance of the model's fits derived from a permutation test (Methods)—we found that the proportion of these cells was higher in LM than in V1 (V1: 58% ± 8% s.e., LM: 78% ± 3%, $p = 1.0 \times 10^{-4}$, paired t-test, $n = 10$), with a higher average explained variance in LM (Fig. 3k; Supplementary Fig. 6e). The energy cross-correlation statistics had the largest contribution to the explained variance (Fig. 3l), which was also confirmed by an analysis of "unique" variance explained[75] (withholding a particular group of PS statistics), and it was found that the energy cross-correlation statistics was again the main contributor (Fig. 3m).

As these results show, underlying the increased selectivity for textures in area LM and a larger proportion of cells having such selectivity is a stronger responsiveness to statistical features that are texture-defining—that is, those quantified by the energy cross-correlation PS statistics.

**Population responses to texture images**

Next, we examined whether we could identify signatures of texture selectivity, more significantly so in LM than in V1 at the level of population encoding. To discriminate the activity of the texture–scramble pairs in V1 and LM, we trained a binary logistic classifier. The decoder was largely above the chance level (50%) for all pairs (Fig. 4a), with significantly larger performance in LM than in V1 when grouping all the texture families (V1, 77% ± 1% (s.e.); LM, 81% ± 2%, $p = 0.007$, paired t-test, $n = 10$ mice). In both V1 and LM, the rocks family was the one with the lowest classification accuracy (Fig. 4a, $p = 4 \times 10^{-7}$, one-way ANOVA; performance of rocks different from all pairs, repeated measures correction, $p < 0.035$, post-hoc Tukey HSD test, $n = 10$). Notably, a similar drop in performance was also observed in the d′ measures of behavioral performance, where the lowest performance was observed for this texture–scramble pair consistently in individual mice trained across all four texture–scramble pairs, and across animals (Fig. 4b, $p = 3 \times 10^{-5}$, one-way ANOVA; repeated measures correction, $p < 0.03$, post-hoc Tukey HSD test, $n = 16$).

**Linking image statistics to neural and behavioral representations.** Next, we examined whether the correlation in neural and behavioral discriminability could be related to the statistics of the images. For instance, if the statistics of the rock exemplars were particularly similar to those of their scrambles compared to other families, then this reduced statistical discriminability may explain the drop in both behavioral and neural discriminability. We thus defined a distance metric in a statistical stimulus space based on a reduced set of PS statistics (Fig. 3g). In each subspace, we measured the inter-cluster distances (normalized by the clusters' spread) between the textures and the corresponding scrambles, finding that the rocks family had a significantly smaller texture–scramble distance than the other families in the energy statistics subspace (Fig. 4c, e). For the other statistical subspaces, although the texture–scramble distances of rocks were still the shortest compared to the other families, they overlapped with at least one other family (Supplementary Fig. 10).

Together, the correlation between the PS-distance metric in the energy subspace, which best captures texture-defining statistics, and the drop in neural decoding and behavioral performance associated with the rocks family, suggest a tight linking framework between high-

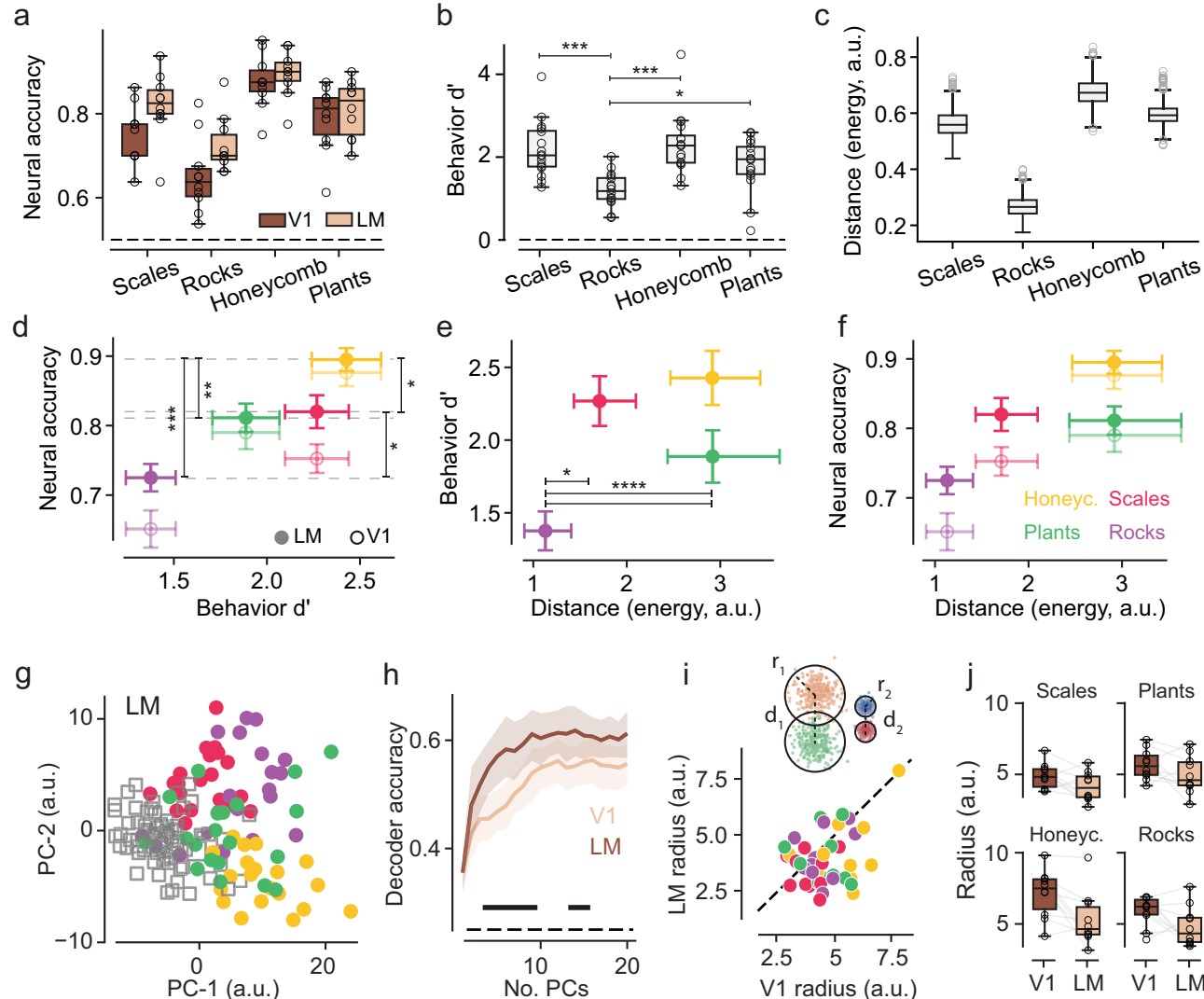

**Fig. 4 | Statistical, behavioral, and neural discriminability correlate with the geometry of texture representations in LM. a** Accuracy of a linear classifier trained to discriminate textures from the scrambles for all four families using the neural responses from LM-ROI and V1-ROI. Each dot indicates a mouse. The decoder accuracy is above 50% chance level for all pairs (V1: scales 75% ± 2% s.e. $p = 1 \times 10^{-6}$, V1: rocks 65% ± 3%, $p = 3.5 \times 10^{-4}$, V1: honeycomb 88% ± 2%. $p = 2 \times 10^{-8}$, V1: plants 79% ± 2%. $p = 1.1 \times 10^{-6}$, LM: scales 82% ± 2%. $p = 2 \times 10^{-7}$, LM: rocks 72% ± 2%. $p = 1.9 \times 10^{-6}$, LM: honeycomb 89% ± 2%. $p = 3.0 \times 10^{-9}$, LM: plants 81% ± 2%. $p = 1.5 \times 10^{-7}$; one-sample $t$-test, $n = 10$ mice; ANOVA values reported in **d**) Box plots indicate the median with a horizontal bar; the box height denotes the inter-quartile range (IQR, 1st and 3rd quartile) and the whiskers extend by 1.5 x IQR. **b** Behavioral discriminability (d'), as in Fig. 1f, but for a subset of the mice that completed the texture–scramble tasks for all four families ($n = 16$, open dots; $p = 2.0 \times 10^{-5}$, one-way ANOVA across families; post-hoc analysis (Tukey HSD): rocks vs. honeycomb: $p < 0.0001$, rocks vs. plants: $p = 0.04$, rocks vs. scales: $p = 0.0002$). Box plots as in $a$. **c** Normalized distances for each texture/scramble family pair for the energy cross-correlation statistics of the images; gray dots indicate outliers from a bootstrapping procedure (Methods). Box plots as in **a**. **d** The combined 2D plot from the data in **a** and **b**; the error bars are s.e.; colors as in **f**; reference gray horizontal broken lines for pairwise statistical comparisons for area LM (filled dots), one-way ANOVA for texture families: $p = 3.7 \times 10^{-5}$; post-hoc analysis (Tukey-HSD): honeycomb-plants: $p = 0.04$; honeycomb-rocks: $p = 1.2 \times 10^{-5}$; plants-rocks: $p = 0.03$; rocks-scales: $p = 0.016$. V1 classifier, one-way ANOVA for texture families (asterisks not shown): $p = 1.7 \times 10^{-6}$; post-hoc analysis: honeycomb-rocks: $p = 6 \times 10^{-7}$; honeycomb-scales: $p = 0.005$; plants-rocks: $p = 0.0013$; rocks-scales: $p = 0.009$; $n = 10$, mean accuracy for each mouse. **e** Behavioral discriminability as in **b** plotted against the inter-cluster distances (**c**) for each texture/scramble family pair and for the energy cross-

correlation statistics. The error bars are s.e. for the behavioral data and bootstrap confidence intervals with Šidák correction for multiple comparisons ($\alpha = 0.05$) for statistical distance: CIs = 99.15, $p_{boot} < 0.05$, for all pairwise comparisons with rocks; $n = 1000$ bootstraps. **f** Neural classifier accuracy, as in **a**, against the inter-cluster distances for the energy cross-correlation statistics (**c**); $n = 10$ mice. **g** 2D scatter plot for the first two PCA components of the neural responses from LM-ROI for one example animal; each dot is an exemplar (averaged across repeats, image rotations, and time frames around the peak response); filled circles for textures and empty squares for scrambles; colors as in **f**. **h** Accuracy of a multinomial classifier ($n = 10$ mice) discriminating between the texture families as a function of the number of components, separately in V1 and LM PCA spaces. The shaded regions correspond to the 95% confidence intervals across all mice; the center is the mean accuracy of the classifier (i.e., the mean of the 5-fold cross validation procedure) across all 10 mice. The black horizontal bars indicate the range of PCA components for which the classifier accuracy is statistically different between V1 and LM (paired t-test, $p$-values < 0.05). **i** Top: schematic plot illustrating the metrics used in the neural PCA space. For every cloud of points in the PCA space, we measure its radius (e.g., $r_1$, $r_2$) and its distance with respect to another cloud (e.g., $d_1$, $d_2$). The clouds on the left show larger radii and inter-cluster distance compared to the clouds on the right. Bottom: scatter plot of the cluster radii in V1 (x-axis) and LM (y-axis) for all mice ($n = 10$). Each dot is a cluster radius for a given texture family and mouse; colors as in **f**. The black dotted line is the diagonal. **j** Radius values in V1 and LM; each dot corresponds to a particular mouse and each panel a different texture family ($n = 40$, 10 mice x 4 families), gray lines are paired animals and families between V1 and LM; $p = 0.04$, repeated-measures ANOVA brain-area effect; $p = 6.0 \times 10^{-6}$ family effect; $p = 0.002$ interaction effect. Box plots as in **a**. Source data are provided as a Source Data file.

order image statistics, population encoding in V1 and LM, and behavioral performance (Fig. 4d–f).

**V1 and LM differences in the representational geometry of texture families.** The results from the binary logistic classifier trained to discriminate between the texture–scramble pairs suggest representational differences between V1 and LM. For instance, significantly fewer principal components (PCs) were needed in LM to attain maximum performance (two to four dimensions) whereas V1 required thrice as many, between four and 12 PCs (Supplementary Fig. 11). Further evidence for representational differences between V1 and LM was provided by a decoding analysis attempting to discriminate between texture families from the neural activity in V1 and LM. We used a multinomial logistic classifier trained to categorize the four texture families across each of the 40 exemplar images for each family. Since the number of cells differed across experiments, we used PCA to fix the representational dimensionality of the activity space. Even with only two PCA components, the collective activations of the visually responsive cells across all texture and scramble stimuli already formed separate activity subspaces (or "clusters", Fig. 4g) with an average explained variance above 15% (V1: 15.5% ± 1.4% s.e. LM: 19.1% ± 1.2%). The cross-validated classifier performed significantly above chance level in both areas, plateauing at approximately 60% performance with ~10 PCA components (Fig. 4h). The LM decoder outperformed the V1 decoder, with significant differences observed reliably in the range between two and sixteen PCA components (Fig. 4h).

To highlight the properties of the population encoding that could explain the increased classification performance in LM, we studied the geometry of texture representations in a shared 16-dimensional PCA space of V1 and LM activations, in which the texture-texture decoder had the largest (significant) discriminability power (Fig. 4h). Each point in this space corresponded to a texture exemplar (averaged across repeats) labeled according to the corresponding texture family (2D schematic of the 16D representations in Fig. 4i). For every family pair (40 exemplars per family) we computed a Mahalanobis distance measure, which demonstrated an overall increase in distance in LM compared to V1, in agreement with the performance of the multinomial classifier (24 ± 5% distance increase in LM vs. V1, $p = 0.002$ paired t-test, $n = 60$, 10 mice x 6 pairs). Both the classifier and the Mahalanobis distance measure are sensitive to the relative "shapes" of the underlying distributions. To test for simple geometrical changes from V1 to LM, for every family we computed the spread of the activations associated with the 40 exemplars—that is, the radii of the activity subspaces and their pairwise distances ("inter-cluster" Euclidean distances). In LM, we found that the cluster radii were significantly smaller than in V1 (Fig. 4i,j; repeated-measures ANOVA: interaction effect, $p = 0.002$; brain-area effect, $p = 0.04$; family effect, $p = 6.0 \times 10^{-6}$), with no evidence for smaller inter-cluster distances in LM compared to V1.

In conclusion, a population-level signature of the increased selectivity for energy cross-correlation statistics in LM is a change in the representational geometry of the texture stimuli with LM having more "compact" representations than V1, as evidenced by the smaller subspace radii. These findings suggest that the more compact representations in LM contribute to the increased classification performance observed in this area.

## Discussion

We found that mice can perceptually detect higher-order statistical dependencies in texture images, discriminating between textures and scrambles and between different texture families. Across visual areas, V1 and LM were those most prominently selective to texture statistics, with LM more so than V1, significantly driven by the energy cross-correlation image statistics. The representational geometry of

population responses demonstrated subspaces for each texture–scramble pair, with better stimulus decoding in LM than in V1. The distances between the texture–scramble subspaces changed according to the stimulus statistical dependencies, more significantly in the energy cross-correlation statistical components. The textures statistically most similar to scrambles (i.e., exemplars from the rocks family) had the shortest distances between the corresponding neural subspaces, with the worst perceptual discriminability by the animals and by a decoder trained on the neural representations. This was observed consistently in the animals trained on various texture–scramble pairs as well as across animals for this specific pair. Finally, the neural representations for different texture families were also easier to discriminate in LM than in V1, with LM having more compact subspaces (smaller radii) for individual textures.

Efficiency, in reference to the efficient coding hypothesis[22], highlights a correspondence between input statistics, perceptual sensitivity, and the allocation of computational (and metabolic) resources. A neural code is efficient if it can reflect environmental statistics; such a code will favor basic visual features that are more common, relying on non-uniform neural representations and percentual sensitivity[23,24,26,27]. This implies a close correspondence between neural, perceptual, and statistical representations. We studied this correspondence by examining the geometry of such representations in V1 and LM and identifying "rocks" as the family most similar to its scramble exemplars, with neural-distance representations and behavioral performance also being the smallest for this family. This was reliably observed in animals (tested across various texture–scramble pairs) and across animals for this pair. The selected texture families were chosen because of their likely ethological relevance to mice (e.g., rocks and plants) and their extensive use and characterization in the texture literature[38,59]. They also had sufficiently diverse statistical dependencies to permit a simple statistical similarity ranking between the texture–scramble pairs. However, future work could adopt a more principled approach in selecting texture families based on the statistical distance measure, as adopted in this study. This would allow us to define a psychometric difficulty axis in the stimulus-statistics space to be explored parametrically, both for texture–scramble and texture–texture discrimination. For the latter in particular, this approach could overcome a current statistical limitation: the six texture-family pairs span a relatively narrow range of distances in stimulus statistics, requiring an extremely large number of trials to test for differences in behavioral performance and neural representations, both within and across mice. Texture synthesis guided by a predetermined sampling of the relevant distances along a psychometric difficulty axis could ease the burden of collecting an exceedingly large dataset.

To examine the perceptual ability of mice to discriminate textures, we carefully controlled the stimulus statistics of each exemplar. We customized a CNN-based approach for texture synthesis to achieve the equalization of lower (e.g., luminance, contrast, and marginal PS statistics) and higher statistical dependencies (e.g., linear and energy cross-features PS statistics). Further, we normalized the power spectrum in a frequency band of high perceptual sensitivity for mice and generated several metameric exemplars[43] differing in pixel-level representations but otherwise having identical statistical dependencies. We also introduced image rotations to ensure that the animals could generalize along this stimulus dimension. Finally, we tested the trained animals with new sets of metameric exemplars, confirming that "brute force" memorization of low-level features was not used in the task[67]. This approach gave us control over which statistical features the mice could use in the task and which component is critical when linking the statistical dependencies of the visual stimuli to neural and perceptual representations. In this respect, our approach may be preferable to using *synthetic* textures, in which typically only a reduced set of statistics of interest is under parametric control, while others are left free to (co)vary[13,14,76–78].

The linking framework between stimulus statistics, neural representations, and perceptual sensitivity was most significant for the energy cross-correlation statistics. These statistics capture dependencies in high-order spatial correlations, sampling different parts of the image with filters having different spatial scales and orientations[15]. Therefore, energy statistics are sensitive to the "ergodic" properties of textures, that is the homogeneity of the statistics across different parts of an image or, for a given part, across exemplars from the same texture family[1]. Spatial structures and patterns repeated across the image (e.g., elongated contour bands) are absent in scramble images, which however are matched in average orientation power to texture images. It is possible that mice rely on these patterns in the behavioral tasks. Previous psychophysical studies in humans[34] have shown that energy components play a crucial role in predicting perceptual sensitivity to texture images, motivating the use of perceptual components in models of texture synthesis[59]. However, bridging the gap between the perceptual strategies employed by mice in our tasks and the link between energy statistics and human perception remains challenging. Finally, additional research is necessary to extend our findings to non-ergodic natural images, such as images of objects and faces, also characterized by high-order spatial correlation properties.

The asymmetry in texture-scramble discriminability (d') observed in V1, with a bias for the upper visual field, could reflect an adaptive mechanism to natural-image statistics. Previous studies have identified gradients in mouse V1 related to variables such as binocular disparity[70], coherent motion[71], and UV-green color contrast[72]. These gradients have been related to statistical properties in visually relevant environments. Likewise, the observed gradient in the discriminability measure (d') may signify a heightened sensitivity to high-order spatial correlations in the upper visual field associated with natural elements in the visual scene—not uniquely textural—such as predators or landmarks used for navigation.

The prominent texture selectivity found in area LM is consistent with what is known about the area specialization of the mouse visual cortex, implicating LM in the processing of content-related (semantic) visual information[74,79–86], in high-fidelity representations of spatial features—including those of textures[58] and with inactivation studies demonstrating the necessity of LM for the perception of even simple visual stimuli[80,83].

At the circuit level, an analysis of the representational geometry of LM population responses[87,88] revealed distinct activity subspaces associated with different texture families. These texture "manifolds" are reminiscent of the concept of object manifolds introduced in relation to the processing of complex objects along the ventral stream in primates[89–92] and in mice[57,93]. When comparing LM to V1 representations, we found a reduction in the size (radius) of texture clusters, with this effect leading to an overall improved linear discriminability of texture families in LM compared to V1. One interpretation is that the increased discriminability from V1 to LM is related to an increase in the representational invariances to image statistics, as suggested by previous studies on rats[55,56] and mice[57]. The reduction in cluster sizes reflects an overall more compact representation of the four texture families, which may relate to LM achieving a higher encoding capacity than V1 while, at the same time, retaining large encoding accuracy for textures. Another possibility is that the V1 texture representations reflect an "incomplete inheritance" from LM via top-down signal processing[94]; experiments inactivating LM while recording from V1 could elucidate this point.

Neural recordings were done in untrained animals passively viewing the stimuli, thus enabling comparisons with primate studies that used similar preparations[34,36,95]. Furthermore, neural recordings in untrained animals eliminate the possibility that the observed selectivity and representational features emerge as a consequence of the task-learning process. Rather, our analyses likely highlight a computational property of the visual system emerging from an evolutionarily refined genetic program[28] and from exposure to a rich set of image statistics during development. The observation that in naïve animals the decoding quality of the neural signals follows the statistical separability of texture–scramble images, mirrored by congruent performance modulations in trained animals, supports this interpretation. It is also conceivable that learning and attentional processes, as animals engage in tasks, might affect the properties of neural representations[1,96,97]. Therefore, in future studies, it would be of interest to examine the neural dynamics underlying texture representations during the different phases of learning.

In conclusion, our results demonstrate the signal processing of naturalistic stimuli in the mouse visual cortex akin to what has been observed in primates, additionally highlighting an intimate link between the geometry of neural representations, stimulus statistical dependencies, and perceptual behavior, which is a distinct hallmark of efficient coding principles of information processing. Considering that similar processing features are also found in V2/LM equivalents in artificial neural networks, our results likely reflect a general efficient coding principle emerging in hierarchically organized computational architectures devoted to the extraction of semantic information from the visual scene.

## Methods

### Subjects
All procedures were reviewed and approved by the Animal Care and Use Committees of the RIKEN Center for Brain Science. The behavioral data for the texture–scramble and texture–texture discrimination visual task were collected from a total of 21 mice: six CamktTA;-TREGCaMP6s (four males and two females), 14 C57BL/6 J WT (11 males, three females), and one male CaMKIIα-Cre. For the passive imaging experiments, we used a total of 11 mice (11 for widefield and 10 for two-photon): six CaMKIIα-Cre transgenic mice (four males and two females) and five C57BL/6 J WT (two males and three females). The age of the animals typically ranged between eight and 28 weeks old from the beginning to the end of the experiments. The mice were housed under a 12–12 h light–dark cycle. Temperature was kept in the 20-24 °C range and humidity at 45-60%.

### Cranial window implantation
As described in ref. 64, for the implantation of a head-post and optical chamber, the animals were anesthetized with gas anesthesia (Iso-flurane 1.5–2.5%; Pfizer) and injected with an antibiotic (Baytril, 0.5 ml, 2%; Bayer Yakuhin), a steroidal anti-inflammatory drug (Dex-amethasone; Kyoritsu Seiyaku), an anti-edema agent (Glyceol, 100 µl; Chugai Pharmaceutical) to reduce brain swelling, and a painkiller (Lepetan, Otsuka Pharmaceutical). The scalp and periosteum were retracted, exposing the skull, and then a 5.5 mm diameter trephination was made with a micro drill (Meisinger LLC). Two 5 mm coverslips (120-170 µm thickness) were positioned in the center of the craniotomy in direct contact with the meninges, topped by a 6 mm diameter coverslip with the same thickness. When needed, Gelform (Pfizer) was applied around the 5 mm coverslip to stop any bleeding. The 6-mm coverslip was fixed to the bone with cyanoacrylic glue (Aron Alpha, Toagosei). A round metal chamber (7.1 mm diameter) combined with a head-post was centered on the craniotomy and cemented to the bone with dental adhesive (Super-Bond C&B, Sun Medical), which was mixed with a black dye for improved light absorbance during imaging.

### Viral injections
For imaging experiments, we injected the viral vector rAAV1-syn-jGCaMP7f-WPRE ($4 \times 10^{12}$ gc/ml, 1000 nl) into the mice's right visual cortex (AP, −3.3 mm: LM 2.4 mm from the bregma) at a flow rate of 50 nl/min using a Nanoject II (Drummond Scientific, Broomall, Pennsylvania, USA). The injection depth was 400 µm. After confirmation of

fluorescent protein expression (approximately two weeks after the AAV injection), we made a craniotomy (5.5 mm diameter) centered on the injection site while keeping the dura membrane intact and implanted a cover-glass window, as described above.

## Behavior

**Behavioral training procedure.** Water-restricted mice were habituated to our automated behavioral training setups with self-head fixation, as previously described[64]. The training of mice progressed according to four stages with increasing difficulty, both procedural and perceptual, and with the fourth stage involving the final tasks described in the Results section (both the texture–scramble and texture–texture tasks). In the first stage, trial timing and stimulus properties were already set as in the final stage (Fig. 1d). However, 1) the "go" stimuli were shown in 70% of the trials (instead of 50% as in the fourth stage); 2) the minimum wheel rotation required to trigger a response was 5° instead of 45°; 3) the maximum wheel rotation that was allowed during the last second of the ITI was larger (20° instead of 5°); 4) the reward size was 8 μl instead of 4 μl. During this training stage, mice learned the association between wheel rotation and water reward contingent on the stimulus presentation on the screen. After they learned to rotate the wheel contingent to stimulus presentation in at least 80% of the trials for three consecutive sessions, they were moved to the second training stage, with the following changes: (1) the "go" stimuli were shown in 70% of the trials; (2) the wheel rotation angle to signal a response was increased to 15°; (3) the maximum wheel rotation allowed during the ITI was decreased to 5°; and (4) the water reward was lowered to 4 μl. After the mice reached at least 70% hits for three consecutive sessions, they were moved to the third training stage, in which the only change was an increase in the wheel rotation angle to 30° to signal a response. After reaching at least 70% hits for three consecutive sessions, the mice were moved to the fourth and final training stage with 50% hit trials. Most of the mice started the training with the honeycomb or scales texture/scramble family. Afterwards, we randomly selected the next family until all four families were successfully discriminated against the corresponding scrambles. A texture–scramble family discrimination was considered completed when the mouse had a d' > 1 consistently over 10 consecutive sessions. The training details for the texture–texture task are described in the "Texture–texture task" section. In the final stage, mice received 4–5 ml of water daily. In preceding stages, in cases where mice failed to acquire sufficient hydration to maintain a healthy body weight (measured daily), specifically falling below 75% of the baseline body weight, we administered a significant bolus of water gel after the session. Alternatively, we temporarily withdrew the animal from the training protocol in instances where the mouse struggled to rapidly regain a healthy weight. Mice were typically not trained on weekends when they had free access to water.

**Behavioral performance.** We evaluated behavioral performance using the discrimination metric d-prime (d') from signal detection theory[98] which is defined as: d' = Z(hit-rate) - Z(false-alarm-rate), where Z is the inverse cumulative normal distribution function, "hit rate" is the proportion of correct "go" trials, while "false alarms" is the proportion of "no go" trials with erroneous responses.

**Texture–scramble task.** Mice independently fixed their headplate to the latching device twice a day in a fully automated behavioral setup[64] that was connected to their home-cage. It comprised a self-latching stage, a rubber wheel with a quadrature encoder sensor to read the wheel's position[99], a spout that dispensed water drops (4 μl), and a computer monitor positioned in front of the latching stage. Mice were required to rotate the toy wheel with their front paws contingent on a texture stimulus shown on the screen (the "hit" trials were rewarded with a water drop; the "false alarm" responses were discouraged by presenting a full-field flickering checkerboard pattern for 10 seconds; no feedback was given for "misses" and "correct rejects"). Regarding the temporal structure of the trial (shown in Fig. 1d), a session began with an ITI with an isoluminant gray screen (with the same mean luminance level of the texture and scramble images). The ITIs lasted for four to six seconds chosen from a randomly uniform distribution. Mice had to refrain from rotating the wheel, with movements during a one-second period before the onset of the visual stimulus extending the ITI by one second. The stimuli had a 50% chance of being either a go stimulus (texture exemplar) or a no-go stimulus (scramble exemplar). The parameters of the stimuli matched those used in the imaging experiments: 100° in visual angle, with a raised cosine mask to reduce sharp edges (high-frequency components), and the texture family to be discriminated was kept constant during the entire session, randomly selecting the image to be displayed in each trial from a set of 20 exemplars. Following the stimulus presentation, the mice had two seconds to respond (response window). A wheel rotation was counted as a response if it exceeded 45°. After a hit trial, a water reward was given, which was followed by a one-second period, during which the stimulus remained visible on the screen, which then disappeared at the beginning of the ITI period with a randomized four to six second duration. In false-alarm trials, the stimulus disappeared after the wheel rotation, and a flickering checkerboard pattern (2 Hz) was displayed for 10 seconds followed by an ITI period. For miss trials, a new ITI began at the end of the two-second response window. The session ended either when the mice received 400 μl of water or when the session's duration reached 1800 seconds. To verify that the mice did not rely on "brute force" memorization of the luminance patterns shown on the screen to solve the task[100], in a subset of expert animals (n = 17), trained on all four texture–scramble family pairs, we introduced new sets of texture and scramble exemplars (20 each) and compared the performance of mice in the five sessions before and after the change in exemplars.

**Texture–texture task.** The mice trained in the texture–texture go/no go task were both a subset of the mice trained in the texture–scramble (n = 14) and a new cohort of naïve mice (n = 2). If the mouse had been previously trained in the texture–scramble task expert, we simply changed the protocol so that a randomly chosen texture family (20 exemplars) was the new "go" stimuli and, similarly, another randomly chosen texture family (20 exemplars) was the new "no go" stimuli. Instead, for naïve mice, we trained them following the same training procedure described for the texture–scramble task but using exemplars from another randomly chosen texture family instead of scrambles.

## Image synthesis

**Texture synthesis.** As described in ref. 59, convolutional neural networks (CNNs) can be used to extract a compact representation of texture images by measuring the activation patterns of a CNN to a given texture. These activations are an over complete multi-scale representation[59,101] that can be used to synthesize an arbitrary set of texture exemplars. Specifically, the first step for the synthesis of a novel texture exemplar relative to a reference texture ("target", x) is to obtain a CNN parametrization of x—that is, its feature vector representation, f(x). This is done by concatenating the spatial means of the feature-map activations in each of the five VGG16 layers, which results in a feature vector of size 1,472: $f(x) = \{\mu_{\tilde{x}_j}^{(i)}; i = 1, \dots, m; j = 1, \dots, n_i\}$, where $m = 5$ is the number of convolutional layers, $n_i$ is the number of feature maps in convolutional layer $i$, $\tilde{x}_j$ is the spatial mean across filter activations in the feature map $j$, and $\mu_{\tilde{x}_j}^{(i)}$ is the set of such means for layer $i$. The second step is to obtain a feature vector representation, $f(y)$, of a Gaussian-noise image $y$:

$f(y) = \{\mu_{y_j}^{(i)}; i = 1, \ldots, m; j = 1, \ldots, n_i\}$. To obtain $f(x) \approx f(y)$, we solve an optimization problem (with an L1 loss):

$$y^* = argmin_y \Sigma |f(x) - f(y)| \qquad (1)$$

Where $y^*$ is the fully optimized image relative to the target image, $x$.

This approach is nearly identical to that of ref. 59, with the only difference being that we did not add the mean of the three-color channels of $x$ to our feature transform[59] since in our framework it created some degree of "pixelation" in the synthesized images. Rather, as an additional step after optimization, we normalized the images to have equal mean luminance and standard deviation (RMS contrast), as detailed below.

For the synthesis of the textures in Supplementary Fig. 7 we used the Portilla-Simoncelli algorithm to generate images for four texture families. First, we computed the PS statistics of four original textures (using randomly chosen exemplars), and then we generated new textures constraining linear, marginal, and spectral statistics to be those of the "scales" family. Instead, the energy statistics were the original statistics of each of the four texture families (MATLAB function "TextureSynthesis.m", http://www.cns.nyu.edu/~lcv/texture/).

**Texture normalization.** To ensure that texture exemplars had the same lower-order statistics (mean luminance and RMS contrast), we z-scored the pixel intensity values, multiplied them by a fixed contrast (standard deviation, $\sigma = 0.15$), and, finally, added a fixed mean luminance value ($\mu = 0.5$). This normalization was applied to all the "target" texture images (relative to the synthesis procedure with VGG16) and the synthetized texture exemplars, as there were small differences in the luminance and contrast relative to the target after each exemplar was synthetized. Furthermore, to ensure that the spatial frequency content of the textures was within the range of mouse perceptual sensitivity, we used an iterative algorithm in which we progressively rescaled the "target" texture images such that 1) > 95% of the spatial-frequency amplitudes of all the target textures lied within the 0.0 to 0.5 cpd[102] interval; 2) the average amplitude spectrum overlapped across families in the frequency range between 0.01 and 0.5 cpd.

**Scramble generation.** Scrambles are the noise images spectrally matched to the textures[34] generated via FFT-transform of a given texture exemplar (changing for different texture–scramble pairs) and randomizing the phase components while keeping the amplitude ones. Phase randomization was done by drawing the phase values from an FFT-transform of a Gaussian-noise image. The thus-generated scrambles retained the same average orientation and spatial-frequency power as the texture exemplars but lacked the higher-order statistical dependence of the textures[34]. For each of the synthesized scrambled images, we verified that the mean luminance and RMS contrast remained nearly identical to the original textures. The difference was within the floating-point error.

**Images with similar skewness and kurtosis.** We generated a new set of texture and scramble images in which we normalized the luminance histograms (scikit-image function 'match_histograms'). This function was applied before the rescaling procedure (see "Texture normalization"), therefore skewness and kurtosis values were not exactly equal between images after rescaling; however, by being randomly mixed, skewness and kurtosis could not be used to separate between textures families and textures from scrambles (Supplementary Fig. 3a). We then created a texture-scramble and texture-texture discrimination task in which mice used these new images (n = 7 mice, newly trained). These mice had to discriminate between "scales" images and corresponding scrambles, and between "scales" and "plants" textures.

## Image analysis

**Image statistics.** We explored the image statistics at various levels of complexity. Our texture normalization procedure ensured that the pixel histogram distributions had identical means and standard deviations between the images (i.e., luminance, and RMS contrast). Within families and between a matching pair of textures and scrambles, we also confirmed that the average orientation and spatial frequency content were the same. To do so, in each image Fourier transform, we measured the average power in "slices" of the spatial frequencies and orientations (spatial frequency bins: [0.01 cpd, step 0.02 cpd: 0.5cpd]; orientation bins: [0°: step 15°: 180°]). The plots in Supplementary Fig. 1b, c show the amplitude values as a function of spatial frequencies and orientations, averaged across 20 exemplars for all families and stimulus types, and normalized to 1. To measure the higher-order statistics of the images, we decomposed them using an approach devised by Portilla and Simoncelli[15], which decomposes an image using a bank of linear and energy filters tuned to different orientations, spatial frequencies, and spatial positions. The correlations are then computed across the outputs of these filters (i.e., the "PS statistics"). The parameters and classification of the PS statistics we adopted follow what has been previously described[34,36,38,39]. Briefly, we used a filter bank composed of four spatial scales (four downscaling octaves), four orientations (0°, 45°, 90°, 135°), and a spatial neighborhood of seven pixels to compute the filter output correlations. In addition, the marginal statistics of the pixel distributions were also computed (min, max, mean, standard deviation, skewness, and kurtosis). However, since part of our image synthesis pipeline procedure already ensured equal mean and standard deviation, only the differences in skewness and kurtosis were added to the characterization of the image statistics. In the end, the output of this image decomposition yielded four main groups of PS statistics: 1) marginal statistics (skewness and kurtosis); 2) spectral statistics; 3) linear cross-correlation statistics; and 4) energy cross-correlation statistics.

**Dimensionality reduction of PS statistics.** The number of parameters associated with the PS decomposition is larger (740) than the total number of images (320, eight image categories—four texture families and four scrambles—and 20 exemplars per category with two rotations). We thus reduced the number of parameters by applying Principal Component Analysis (PCA) to each PS statistical group after z-scoring the parameter values. We retained at most eight components in each group, which explained at least 70% of the variance per group (Supplementary Fig. 8). The marginal statistics with only two "dimensions" were excluded from this decomposition. After PCA, we again z-scored the outputs across exemplars to ensure that the range of parameter values between the groups of statistics was commensurate; this was necessary to gain interpretability of the distance metric later introduced, which was based on these reduced PS statistics. We also confirmed that the reduced PS statistics retained sufficient information to discriminate between textures and scrambles, with the energy cross-correlation statistics maximally distinguishing between them (Supplementary Fig. 9a,b).

## Imaging experiments

**Visual stimuli.** The visual stimuli were shown on a gamma-corrected monitor (widefield: IIYAMA Prolite LE4041UHS 40", two-photon: IIYAMA Prolite B2776HDS-B1 27"). Except for the experiments for the discriminability gradient (see related section below), the size of the stimuli was 100° of visual angle with a raised cosine window for vignetting to correct for sharp edges; the stimuli were shown in front of the mouse perpendicular to its midline, which pointed to the center of the screen. The animal was at a distance of ~33 cm from the monitor for widefield experiments and ~24 cm for two-photon experiments. For widefield recordings, the stimuli were presented for 250 ms, followed by 750 ms of an isoluminant gray screen (ITI) before a new trial started.

Each mouse was shown 20 exemplars of four texture families and four scramble families (computed from the textures), a total of 10 times each exemplar, with 200 blank trials (i.e., trials with an isoluminant gray screen and no stimuli). This resulted in a total of 1600 trials with images and 200 trials with no stimulus (blanks). The presentation of each image/blank was fully randomized across the entire session.

The two-photon experiments followed the same temporal structure as the widefield experiments; however, we reduced the number of repeats and added image rotations. Specifically, each mouse was shown 20 exemplars of four texture families and four scramble families: a total of eight times for each exemplar, and two rotations (0° and 90°) of each exemplar, with 160 blank trials. This resulted in a total of 2560 trials with images and 160 trials with no stimuli (blanks). We also recorded the responses to oriented gratings: 100 degrees in size, four orientations (0°, 45°, 90°, 135°), five spatial frequencies (0.02, 0.04, 0.1, 0.2, 0.5 cpd) and 15 repeats per stimulus.

**Widefield imaging.** As described in ref. 103, awake mice were head-fixed and placed under a dual cube THT macroscope (Brainvision Inc.) for widefield imaging in tandem-lens epifluorescence configuration using two AF NIKKOR 50 mm f/1.4D lenses. We imaged the jGCaMP7f fluorescence signals using interleaved shutter-controlled blue and violet LEDs with a CMOS camera (PCO Edge 5.5) with an acquisition framerate of 60 Hz. This dual color recording method ensured that we could capture both the calcium-dependent GCaMP signal (blue light path) as well as the hemodynamic-dependent signal (violet light path), as previously reported in other studies[69]. The blue light path consisted of a 465 nm centered LED (LEX-2, Brainvision Inc.), a 475 nm bandpass filter (Edmund Optics BP 475 × 25 nm OD4 ø = 50 mm), and two dichroic mirrors with 506 and 458 nm cutoff frequencies, respectively (Semrock FF506-Di03 50 × 70 mm, FF458-DFi02 50 × 70 mm). The violet path consisted of a 405 nm centered LED (Thorlabs M405L2 and LEDD1B driver), a 425 nm bandpass filter (Edmund Options BP 425 × 25 mm OD4 ø = 25 mm), a collimator (Thorlabs COP5-A), and joined the blue LED path at the second dichroic mirror. The fluorescence light path traveled through the two dichroic mirrors (458 and 506 nm, respectively) and a 525 nm bandpass filter (Edmund Optics, BP 525 × 25 nm OD4 ø = 50 mm) and was finally captured with the PCO Edge 5.5 CMOS camera using the cameralink interface. Camera acquisition was synchronized to the LED illumination via a custom Arduino-controlled software. The frame exposure lasted 12 ms, starting 2 ms after opening each LED shutter to allow the LED illumination to stabilize. In a subset of the widefield experiments we displayed texture and scramble stimuli on a large screen, IIYAMA Prolite LE4041UHS 40" monitor. Mice were placed 22 cm away from the monitor, with the body midline pointing at the right edge of the monitor. With these parameters the visual stimuli subtended approximately an azimuth range of [−62.4°, +62.4°] and an elevation range of [−48.5°, +48.5°].

**Preprocessing the widefield data.** Data preprocessing was done with custom Python and MATLAB code, with subsequent analyses done in Python. The continuously acquired imaging data were split into blue and violet channels. Then, as described in refs. 103,104, we corrected for the "hemodynamic component" by removing a calcium-independent component from the recorded signal. For every pixel, the blue and violet data were independently transformed into a relative fluorescence signal, $\frac{\Delta F}{F} = (F - aF - b)/b$, where $F$ is the original data, and the $a$ and $b$ coefficients are obtained by linear fitting each time series, i.e., $F(t) \sim at - b$. Afterwards, for each pixel, the violet $\frac{\Delta F}{F}$ signal was low-pass filtered (6th order IIR filter with cutoff at 5 Hz) and linearly fitted to the blue $\frac{\Delta F}{F}$ signal: the hemodynamic-corrected $\frac{\Delta F}{F}$ signal was obtained as $\frac{\Delta F}{F} corr = \frac{\Delta F}{F} blue - (c \frac{\Delta F}{F} violet + d)$, where $c$ and $d$ are the coefficients from linearly fitting the low-pass filtered $\frac{\Delta F}{F} violet$ to the $\frac{\Delta F}{F} blue$ signal, i.e., $\frac{\Delta F}{F} blue(t) \sim c \frac{\Delta F}{F} violet(t) - d$. The continuously acquired data was then split into trial periods comprising

sequences of frames in a temporal window of [−500, +1000] ms relative to stimulus onset. This resulted in a tensor with seven dimensions: [stimulus type (texture or scramble), family type (4), exemplars (20), repeats (10), no. pixels X (256), no. pixels Y (230), no. frames]. Next, we averaged across repeats to obtain an "exemplar response tensor."

**Retinotopy maps.** After the mice recovered from the cranial-window surgery (typically 3 to 4 days), we performed widefield imaging recordings during visual stimulation with counterphase flickering bars to obtain maps of retinotopy. We used a standard frequency-based method[105] with slowly moving horizontal and vertical flickering bars and corrections for spherical projections[74]. Visual area segmentation was performed based on azimuth and elevation gradient inversions[73]. The retinotopic maps were derived under light anesthesia (Isoflurane) with the animal midline pointing to the right edge of the monitor (IIYAMA Prolite LE4041UHS 40"), centered relative to the monitor height, and with the animal's left eye at ~25 cm from the center of the screen.

**Two-photon imaging.** As described in ref. 64, imaging experiments were performed using the two-photon imaging mode of the multi-photon confocal microscope (Model A1RMP, Nikon, Japan) with a Ti:sapphire laser (80 MHz, Coherent©, Chameleon Vision II). The microscope was controlled using the A1 software (Nikon). The objective was a 16x water immersion lens (NA, 0.8; working distance, 3 mm; Nikon). The field of view (512 × 512 pixels) was 532 μm × 532 μm. jGCaMP7f was excited at 920 nm, and the laser power was ~40 mW. Images were acquired continuously at a 30 Hz frame rate using a resonant scanner. To align the two-photon field of view with the maps of retinotopy, we captured a vascular image at the surface of the cortex and used it for reference.

**Preprocessing of two-photon data.** All the analyses, except for neuronal segmentation, were conducted using a custom code written in Python. Cells were segmented using Suite2p[106], followed by the manual classification of the segmented ROIs. We then computed the ΔF/F response values (%) for each neuron by first applying a neuropil correction: $Fc = Fs - 0.7 \times Fn$, where $Fc$ is the corrected signal, $Fs$ is the soma fluorescence, and $Fn$ is the neuropil fluorescence. Then, we computed a baseline-fluorescence value ($F\mu$) as the mean of $Fc$ during the first five seconds of the recordings when no stimuli were shown on the screen. We then detrended $Fc$ (Scipy function scipy.signal.detrend) to remove the slow decrease in fluorescence sometimes observed across several tens of minutes and used the zero-mean detrended signal $Fd$ to compute $\Delta F/F = Fd/F\mu$.

**Data analysis: widefield**

**Defining regions of interest.** For every visual area, we defined a visually responsive ROI (or stimulus ROI) based on the maps of azimuth and elevation obtained from widefield imaging, so as to include a range of [+30°, −10°] in azimuth (relative to the contra- and ipsilateral visual fields, respectively) and elevation (±30°), which, for the azimuth, was a conservative estimate of the retinotopic representation of the stimuli (of size ±50° in azimuth and elevation).

**Peak-response and p-value maps.** Widefield responses to textures and scrambles (Fig. 2b) were computed by averaging across repeats, exemplars, and families; the frames were then averaged in a time widow [200, 400] ms after the stimulus onset, approximately centered around the time of peak response. The temporal response curves in V1 and LM to the textures and scrambles (Fig. 2c) were computed by averaging across repeats, families, and pixels within the response ROIs in V1 and LM; the variability was across the exemplars. The response ROIs were defined based on retinotopy as the cortical region that "mapped" the stimulus location in the visual space. The error bands

indicated a 95% confidence interval across the exemplars. To evaluate the significance of the differential response to the textures and scrambles (Fig. 2f), we tested against a distribution of pre-stimulus responses. Specifically, we first computed the response–difference distributions by subtracting the responses to texture exemplars (averaged across repeats) from the randomly paired scramble exemplars. As before, the frames were also averaged around the time of the peak response, [200, 400] ms after the stimulus onset. This resulted in a tensor with four dimensions: [family type (4), exemplars (20), no. pixels X (256), no. pixels Y (230), no. frames]. By grouping the responses to all the families and exemplars, we generated response-difference distributions for each pixel, each containing 80 data points. We applied the same procedure in a temporal window [−350, −100] ms prior to stimulus onset to obtain "null" distributions for each pixel. Finally, we tested for statistical differences between the pre-and post-stimulus onset distributions using a paired t-test and reporting the associated p-values. This procedure was applied to each animal, and the p-value maps were then used to compute the texture modulation of each visual area.

**Texture-scramble discriminability gradient in V1.** We used full field stimuli (azimuth [−62.4°, +62.4°], elevation [−48.5°, +48.5°]), sufficiently large to activate the entire V1. For each V1 camera pixel, we averaged responses across repeats, and time frames within a window of [200, 400] ms after stimulus onset. Then, we compared average discriminability values ($d'$, textures vs. scrambles) in the upper visual field ([0°, +40°] elevation, [−20°, +20°] in azimuth) vs. the lower visual field ([−40°, 0°] elevation, [−20°, +20°] in azimuth), and tested for a significant difference in average d' values using a t-test. Similarly, we measured the average discriminability values in the left visual field (, [−40°, 0°] in azimuth, [−20°, +20°] elevation) vs. the right visual field ([0°, +40°] in azimuth, [−20°, +20°] elevation). Statistics were computed across 8 recording sessions ($n = 4$ mice, 2 sessions each).

**Texture selectivity of visual areas.** To determine how significantly a visual area was modulated by textures compared to scrambles, we computed the proportion of the significantly modulated pixels ($p < 0.01$, from the p-value maps) within the stimulus ROI of each area (described in the section "Defining regions of interest"). This was separately computed in five visual areas (V1, LM, RL, AM, and PM) that were reliably segmented in all animals (Fig. 2g).

**Texture discriminability.** To compute the texture–scramble discriminability values for V1 and LM (Fig. 2h), we considered the responses to exemplars—separately for textures and scrambles—averaged across (i) repeats, (ii) pixels within stimulus ROIs (see section "Defining regions of interest"), and (iii) time frames within a window of [200, 400] ms after the stimulus onset. We then calculated a texture–scramble discriminability index (d') as follows:

$$d' = \frac{\mu_{tex} - \mu_{sc}}{\sqrt{\frac{1}{2}\left(\sigma_{tex}^2 + \sigma_{sc}^2\right)}} \qquad (2)$$

Where $\mu_{tex}$ and $\mu_{sc}$ are the mean responses to the texture and scramble exemplars (80), and $\sigma_{tex}^2$ and $\sigma_{sc}^2$ the corresponding variances. To calculate the "null distribution" of the d' values shown in Fig. 2h (gray band), we followed the same procedure as above in a time window [−300, −0] ms prior to the stimulus onset, reporting the 5% and 95% percentiles of that distribution.

**Data analysis: two-photon**
**Stimulus-responsive cells.** In a typical experiment, we could segment ~200–450 cells (as described in the section "Two-photon imaging"). To establish whether a cell was visually responsive, in each trial ([−500, +1000] ms relative to stimulus onset) we "frame-zero" corrected ΔF/F

by subtracting the average activity within a pre-stimulus period of [−500, 0] ms. Then, we used a d' discriminability measure (similar to refs. 74,107) by comparing the responses to visual stimuli and to "blanks." Specifically, in each trial and for every segmented cell, we averaged the responses in a window of [250, 500] ms post stimulus onset. We then used these average values to generate two distributions: one from the trials with visual stimuli, the other from the "blank" trials. The distributions with the visual stimuli were computed separately for the individual texture and scramble exemplars and considering the response variability across repeats. For each stimulus exemplar, we then computed a discriminability measure, $d'_{stim}$, as done in refs. 74,107:

$$d'_{stim} = \frac{\mu_{stim} - \mu_{blank}}{\sigma_{stim} + \sigma_{blank}} \qquad (3)$$

Where, $\mu_{stim}$ is the mean response across repeats for the chosen exemplar, $\mu_{blank}$ is the mean response across the repeats of blank trials, and $\sigma_{stim}$, $\sigma_{blank}$ are the corresponding standard deviations. This procedure generated a distribution of $d'_{stim}$ values for each cell. A cell was considered visually responsive if the maximum value of this distribution was ≥ 1, and if ΔF/F ≥ 6% in the stimulus-response window (for consistency with refs. 74,107). Subsequent analyses were performed on this subset of stimulus-responsive cells.

**Texture–scramble d-prime.** For every stimulus-responding cell, we considered frame-zero corrected ΔF/F data, averaging across repeats and responses in a time window of [250, 500] ms after stimulus onset. We then considered the data variability across exemplars (and their rotations) to compute a discriminability measure d' as follows:

$$d' = \frac{\mu_{tex} - \mu_{sc}}{\sqrt{\frac{1}{2}\left(\sigma_{tex}^2 + \sigma_{sc}^2\right)}} \qquad (4)$$

Where $\mu_{tex}$ and $\mu_{sc}$ are the mean responses to the texture and scramble exemplars, and $\sigma_{tex}^2$ and $\sigma_{sc}^2$ the corresponding variances.

**Regressive model.** Using a set of reduced PS statistics as regressors (see section "Image statistics"), we constructed a linear regressive model (ridge regularized) to predict individual cell responses. For each exemplar, we computed an average response value as the mean ΔF/F (averaged across repeats and frame-zero corrected) in a time window of [250, 500] ms post stimulus onset. For each neuron $i$, the model was trained to capture the responses to different exemplars using the following loss function:

$$\min_{w_i} \left|\left|y_i - Xw_i\right|\right|_2^2 + \lambda\left|\left|w_i\right|\right|_2^2 \qquad (5)$$

Where $w_i$ are the optimization weights, $y_i$ the data, $\lambda$ a regularization parameter, and $X$ the reduced PS statistics (two dimensions per group, i.e., the first two PCs). We confirmed that the model did not perform significantly better when using more PCs. The model was trained with five-fold cross validation to reduce overfitting, and the regularization parameter $\lambda$ was optimized using a grid search. The model's performance was evaluated in terms of the explained variance (EV) in the cross-validated data. To establish the significance of the model's fit and to derive an EV threshold value for the inclusion of cells in the analyses of Fig. 3l, we used a permutation test. For a given cell, we refitted the responses using as input statistics those from randomly chosen images (across exemplars from all textures and scrambles). Therefore, for each experiment, we obtained a shuffled distribution of EVs (across cells) and chose the 95th percentile of the distribution as the threshold value for significance ($\alpha = 0.05$). We used this approach in all $n = 20$ experiments, resulting in an average threshold value,

$EV_{th} = 0.87\% \pm 0.07\%$ (s.e.). We set a conservative inclusion threshold at $EV_{th} = 1\%$.

**Regressive model: weight analysis.** To examine the contribution of the different reduced PS statistics in the regressive model, we summed the absolute values of the regressive weights separately for each of the four statistical groups: for a given cell, and for the PS group $i$, we computed $W_i = \sum_{j=1}^{d} |w_{i,j}|$, with d = 2, that is, the number of PCs for the reduced PS statistics. We then averaged $W_i$ across all the cells in a given animal (individual data points in Fig. 3l).

**Regressive model: unique EV.** To examine the unique contribution to the explained variance by the different reduced PS statistics, we measured the loss in EV when training models without a particular statistical group. Specifically, considering a subset of cells with significant explained variance (EV > 10%), we first trained a model with all four groups of PS statistics (full model). Further, we trained four more models, each missing one of the four PS groups. We then computed a measure of unique variance explained, $\Delta EV_{u_i}$, as follows:

$$\Delta EV_{u_i} = 100 \frac{EV_f - EV_i}{EV_f} \forall i \in \{PS_1, \ldots, PS_4\}. \quad (6)$$

Where $EV_i$ is the explained variance of a model trained without the PS group $i$, and $EV_f$ is the explained variance of the full model.

**PCA embedding of neural responses.** For every stimulus-responding cell, we considered the frame-zero corrected $\Delta F/F$ data, averaging across repeats and time frames in a time window of [250, 500] ms post stimulus onset. After z-scoring the responses of each cell to different exemplars, we applied PCA (n = 20 PCs, separately for V1 and LM populations) to "standardize" the population size, thus facilitating a comparison between experiments, each having a different number of segmented cells. An example of a PCA space of neural activity is shown in Fig. 4g for LM recordings (n = 2 PCs).

**Decoding responses to textures and scrambles.** In the PCA spaces of neural activity for V1 and LM, as described in the section above, we considered responses to exemplars separately for each of the four texture–scramble families. For each family, we trained a binary logistic classifier to distinguish texture exemplars from scramble exemplars. The model was five-fold cross-validated, and its performance was evaluated using the average accuracy across the five folds. We repeated the same analysis by varying the number of PCs and examining the related changes in classification accuracy separately for the V1 and LM data (Supplementary Fig. 11a–c). Instead, for the analysis in Supplementary Fig. 9a, a binary classifier was trained to discriminate between texture and scramble images (across all families and exemplars), separately on different PS statistical groups.

**Distance metrics for stimulus statistics.** For each of the four PS statistical groups, we considered a 2D-PCA space of image statistics (see section "PCA of PS statistics"), with two PCs already sufficient for near-optimal classification performance (Supplementary Fig. 9a). The overall distance patterns described in Fig. 4 were consistent when using larger numbers of PCs. A single point in each PCA space corresponds to the statistical representation of an exemplar image based on the associated PS statistical decomposition (reduced to four main PS statistical groups). To compute the radius of a cloud of points (20 exemplar points) for a given family, we computed the standard deviation of the $x$ and $y$ coordinates, $\sigma_x, \sigma_y$, and defined the radius as their mean value $r_i = \frac{\sigma_x + \sigma_y}{2}$. For the inter-cluster distance of a given pair of clouds (i.e., exemplars of textures or scrambles), we first computed the center of mass of the two clouds as the mean of the $x$ and $y$ coordinates, $\mu_x, \mu_y$, measured their Euclidean distance which we then

normalized (divided) by the mean of the two corresponding radii. The inter-cluster distances were calculated for all the matching pairs of texture/scramble families (for Fig. 4c). Further, the radius values were computed for all the families and stimulus types and for all the groups of PS statistics.

**Decoding the responses to texture families.** In the PCA spaces of neural activity for V1 and LM (as described in the section "PCA embedding of neural responses"), we created a linear decoding model trained to classify all four texture families. We used a multinomial logistic classifier with an L1 regularization penalty. The training data consisted of the cells' responses to 160 texture stimuli (four families, 20 exemplars, two rotations). The model was trained using five-fold cross validation, and the regularization factor was optimized with a grid search. The model's performance was evaluated as the cross-validated accuracy averaged across folds. We also examined the dependence of the model's performance on the number of PCs (Fig. 4h).

**Distance metrics for neural representations.** To compare the representational differences between V1 and LM, we created a common PCA space of neuronal activations. For a given mouse, we considered responses to exemplars pre-processed as described in "PCA embedding of neural responses" (before PCA). We then applied PCA to a "concatenated" ensemble of V1 and LM cells to derive a common PCA space with $n = 16$ components. The number of segmented cells and the z-scored response values were commensurate between V1 and LM. Using the PCA projection matrix, and by zeroing responses of the "other" area, we could then separately project the V1 and LM responses in this common space. We then measured the radii of the activation "clouds" in this PCA space for each texture family, as well as the inter-cluster distances for pairs of texture families. To compute the radius of a cloud of points (n = 40 points, 20 exemplars, 2 rotations) for a given family, we computed the standard deviation of the $x$ and $y$ coordinates, $\sigma_x, \sigma_y$, and defined the radius as their mean value $r_i = \frac{\sigma_x + \sigma_y}{2}$. For the inter-cluster distance of a given pair of clouds (i.e., exemplars of two textures), we first computed the center of mass of the two clouds as the mean of the $x$ and $y$ coordinates, $\mu_x, \mu_y$, measured their Euclidean distance. Finally, we compared the radii and inter-cluster distances for all six pairs of families between V1 and LM.

For the Mahalanobis distance analysis, for each animal we considered the clouds of points in the V1-LM shared PCA space and computed a Mahalanobis distance value for each of the six pairs of texture families. An ANOVA statistical test was then performed to quantify the area effect (V1 vs. LM).

**Reporting summary**
Further information on research design is available in the Nature Portfolio Reporting Summary linked to this article.

## Data availability
Data to generate all figure panels is provided as a Source Data file and deposited in https://github.com/CBS-NCB/mouseTextures Source data are provided with this paper.

## Code availability
Analysis code is available at this GitHub repository: https://github.com/CBS-NCB/mouseTextures

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

## Acknowledgements

We thank D. Zoccolan for his feedback on the interpretation of our findings. We thank Yuki Goya, Rie Nishiyama, and Yuka Iwamoto for their support with behavioral training, animal care, and surgeries. We thank O'Hara and Co., Ltd., for their support with the equipment. This work was funded by RIKEN BSI and RIKEN CBS institutional funding, JSPS grants 26290011, 17H06037, and C0219129, and a Fujitsu collaborative grant (to A.B.); RIKEN JRA, University of Tokyo IST-RA, and JSPS-DC2 (to F.B.); HFSP postdoctoral fellowship LT000582/2019 (to J.O.); and SPDR for R.A.

## Author contributions

A.B., J.G., F.B., and J.O. designed the study. A.V.J. facilitated image generation. F.B. created the behavioral protocol, collected all behavioral data, performed all recordings, and conducted all analyses. A.B and F.B. wrote the manuscript. R.A. made helpful comments on the behavioral protocol.

## Competing interests

The authors declare no competing interests.
