## [Peer Review File · Nature Communications]

Efficient coding of natural images in the mouse visual cortexREVIEWER COMMENTS

Reviewer #1 (Remarks to the Author):

In this manuscript, Bolaños et al. investigate texture selectivity in mouse visual cortex, focusing on higher visual area LM. This is an interesting question, as though ventral stream processing has been studied extensively in non-human primates and humans, much less is known about ventral stream processing in the experimentally tractable mouse visual system. The experiments and analysis are generally very well done, though I had a few comments/suggestions that should be addressed. The behavioral results using their automated training system were impressive, and the physiology was clean and analyzed carefully. Specific comments below:

1. There appears to be some anisotropy in the V1 responses to textures (Fig 2b, c, e), with the strongest texture responses in the posterior lateral V1. However, this may be due to the location of the presented stimulus. It would be nice if the authors could test for texture responses across V1 (perhaps using full-field texture/scramble stimuli). Previous work on binocular disparity (La Chioma et al., *Curr Biol*, 2019) and coherent motion (Sit & Goard, *Nat Comm*, 2020) have found retinotopic gradients across V1 and HVAs. It would be interesting to see if similar retinotopic gradients are present in texture responses.
2. I found the results in Fig 4c-f to be interesting but somewhat unconvincing. The analysis ultimately comes down to a correlation measured across four points without any real quantification. Is there a way to improve the statistical rigor of this analysis? For example, would it be possible to perform the same analysis with single texture images instead of texture families to allow more thorough quantification?
3. It would be nice to have an additional section in the Discussion of how the results in this manuscript compare to previous studies of texture/shape responses in lateral visual cortex in rodents from Zoccolan, Tolias, and Smith labs (references 55-58).

Minor typos/corrections:

1. The abstract has two difficult-to-parse sentences (lines 11-18) that should be rewritten for clarity.
2. The sample size reported in the text (line 109) does not match the sample size reported in the figure (Fig 1g).
3. Statistical comparisons in lines 115-116 are not clear.
4. Lines 140, 296: I believe “neural activations” should be “neural activity”

5. The blue/green colors For V1/LM in Fig 3 are quite difficult to distinguish (at least on the printed page). I would recommend changing one of the colors to a more differentiable hue.
6. It is too late to change at this point, but the sample is dominated by males (~70% male). a more equal distribution would be preferable. Exact age range and mean (or median) age should be reported.
7. Typo on the viral titer (line 783). Also, the virus injection volume is very large (~20x typical injection volume). The authors should consider multiple smaller injections or use of transgenic GCaMP mice for widefield imaging.
8. Lines 1006-1008: Typo at end of sentence.
9. Line 1015: Add reference number instead of in line reference.
10. Line 1021: Typo between sentences.
11. Extended Fig 2: Panel d is not described in caption.
12. The statistical reporting is sometimes missing details. For each statistical test, the authors should describe the comparisons groups and report the statistical test, sample size, relevant statistic, exact p-value for all comparisons (often, one or more of these are missing). Similarly, box plots should be described in caption (can be described at end of figure if the same plot type is used throughout, but there is often no description to be found).

Overall, most of these points are fairly minor. This is a very nice paper and will make a nice contribution to the field.

Reviewer #2 (Remarks to the Author):

This study reveals that mice can discriminate families of visual textures, and provides insights into how this ability may derive from activity in two areas in their visual cortex (V1 and LM).

The main findings are that

- (1) Mice discriminate among textures and between textures and statistically simpler matched stimuli. (Fig 1)
- (2) Neurons in V1 and especially LM respond on average more strongly to textures than to the matched stimuli (Fig 2, Fig 3)
- (3) V1 and especially LM encode texture families in subspaces that are further apart when whose families differ more in statistics and are more discriminable perceptually (Fig 4).

These claims seem well supported by the data and the analyses, and the overall work is likely to be of interest to a wide range of scientists interested in high-level vision and mouse vision.

MAJOR

The main suggestion is to explain more clearly what is different across textures and between textures and the scramble controls. A lot of the relevant information is in ED Fig 5. It would be useful to move it to the main text and to use graphics that better segregate the scrambles from the textures. Currently, the figure uses filled circles vs. filled stars, and these don't segregate well. Consider using open vs closed circles or some other notation where the two segregate.

In Fig 4 the paper devotes a lot of attention to the "energy" statistics, but it is not clear that those are so crucial. It is true that those statistics are very different across texture families and between texture vs. scramble (ED Fig 5a). However, this is also true of histogram skewness and kurtosis, the "marginal" statistics (ED Fig 5d). These are much simpler, as they could be computed directly from retinal images. Skewness, in particular, is simply the extreme value of ON and OFF channels. The latter is also called blackshot, and is a powerful discriminant of textures (Chubb et al, 1994, and <https://doi.org/10.1016/j.visres.2004.07.019>, and <https://doi.org/10.1167/3.9.60>).

It would be useful to know if skewness alone could be the cue that distinguishes textures from each other and from the scrambles. This is currently not clear from ED Fig 5d because that figure does not directly show the skewness and the kurtosis. Does that figure need to use PC1 and PC2? It would be so much clearer if it simply plotted skewness vs kurtosis. This would clarify whether the textures can be discriminated based on something as simple as blackshot.

MINOR

In the analysis shown in Fig 1f would be useful to connect the data for the 16 mice that were tested with all families, and use ANOVA or similar to test the hypothesis that some mice are better than others (regardless of texture) and some texture families are easier than others (regardless of mouse). Presumably it would show that the Rocks family was the hardest one. It would also be interesting to know if there is a significant interaction mouse/family.

Reviewer #3 (Remarks to the Author):

The authors present a study examining the processing of textures in the mouse visual system. The study first uses behavioral data showing to show that mice can distinguish textures from phase-scrambled control stimuli as well as between different families of textures. They then use both wide-field calcium imaging and two-photon calcium imaging to show that area LM contains neurons that are particularly sensitive to textures and that LM contains a population code that can distinguish between textures better than V1. They claim that LM represents textures in a more compact population space than V1. In general I think this is a well-executed study which is well-described and has a clear outcome, at least at the behavioral and single-cell levels. The results are a first step to showing the sensitivity of the mouse visual system to texture-stimuli and could in principle open the way to further studies examining the finer details of texture representation in LM and how learning affects these representations. I do have some caveats about the use of phase-scrambled controls and the strength of the claims about a more compact population code, aside from these I think this is a nice study.

1) A major issue with using phase-scrambling to generate control images is that destroys all structure in the image, i.e. it removes all elongated contours from the image. It is therefore hard to determine what the mice were picking upon when discriminating between textures and scrambles – a mouse that has learnt to move the wheel when any elongated contour is present in the image would solve the task. Just visually examining the textures in Figure 1b would seem to confirm my worry as the ‘rocks’ texture has the least amount of high-contrast elongated contours compared to the other texture families and is also the hardest to discriminate. The fact that the animals could also discriminate between texture families is more convincing and suggests that the animals are picking up on texture cues rather than simple structures. But still, I would like to see some acknowledgement and discussion of this issue by the authors.

2) Related to this. It would be useful for a reader to get an intuition about what is being captured by the PS statistics. I would like to see example images which vary along one dimension of this statistical space while keeping the others constant. Perhaps the ‘contour’ dimension I refer to above is what is being captured by the energy dimension? It would in any case be very useful to gain an intuition in how changes along the energy dimension alter the textures

3) I’m sure what to make of Figures 4j-l. The authors report a significant difference in cluster radii and inter-cluster distance between LM and V1 in the text. But in Figure 4l it states that the percentage-wise change is not significant. Given that the ‘compactness’ of the population representation in LM is one of the findings highlighted in the abstract this seems not to be a very robust effect. The scatter-plots also don’t seem very convincing. I would recommend using less strong language concerning these population effects and adding some more nuance to the abstract and discussion of these results. Also – the graphs in Figure 4j-k show multiple points per mouse, how was the paired t-test statistic calculated? With an n of 10 (mice, as reported in the text, did they then average across families?) or an n of 40 (mice x texture family). This is really grouped data with multiple points per mouse and so a repeated measures ANOVA or linear mixed effects model would be more appropriate. As the authors point out, the fact that the radii and inter-cluster distance both decrease could have had counter-acting effects on the performance of the decoder, but this doesn’t appear to be the case. These two measures could be directly combined into a measure such as Mahalanobis distance or a multi-dimensional d-prime which would directly

measure the distance between clusters in units of their standard deviation (it is not 100% clear to me if the inter-cluster distance has been normalized by the radii of the clusters, on line 301 they state that it is the Euclidean distance, but in lines 1179-1184 they talk about dividing by the mean radii, which is correct?). Some clarification is required here.

4) Were the mice water-restricted? Perhaps I missed it, but this procedure should be described in the methods.

Minor points

What is the function of the contours in Figure 4j-k? The scatter-plots would be easier to interpret without them.

Line 29: Should be 'Synthesized'

Line 115: ' $d' = 1.4 \pm 0.14$, $n = 16$, $d' > 1$, $p = 0.016$; $p = 3 \times 10^{-5}$, $n = 15$ ' It isn't clear what the different p-values and n-values refer to.

Line 721: 'One-ay' typo.

Line 734; 'The behavioral discriminability as in (a)' Should be panel (b). Also what re inter-cluster distances?

Line 819: 'Their self-head was fixed' – should be 'their head was self-fixed'.

Line 1008: 'retinotopic' missing word.

REVIEWER COMMENTS

Reviewer #1

In this manuscript, Bolaños et al. investigate texture selectivity in mouse visual cortex, focusing on higher visual area LM. This is an interesting question, as though ventral stream processing has been studied extensively in non-human primates and humans, much less is known about ventral stream processing in the experimentally tractable mouse visual system. The experiments and analysis are generally very well done, though I had a few comments/suggestions that should be addressed. The behavioral results using their automated training system were impressive, and the physiology was clean and analyzed carefully. Specific comments below.

We appreciate the reviewer's recognition of our efforts.

1. There appears to be some anisotropy in the V1 responses to textures (Fig 2b, c, e), with the strongest texture responses in the posterior lateral V1. However, this may be due to the location of the presented stimulus. It would be nice if the authors could test for texture responses across V1 (perhaps using full-field texture/scramble stimuli). Previous work on binocular disparity (La Chioma et al., Curr Biol, 2019) and coherent motion (Sit & Goard, Nat Comm, 2020) have found retinotopic gradients across V1 and HVAs. It would be interesting to see if similar retinotopic gradients are present in texture responses.

Thanks for this observation. Indeed, new analyses have highlighted a clear gradient in V1, with a discriminability measure d' (texture vs scramble) from widefield GCaMP data being significantly higher in posterior V1, corresponding to the upper visual field. To elucidate this point, we performed new experiments using, as suggested, full-field textures and scrambles. We presented the stimuli contra-laterally in a large monitor (40"), which covered approximately an azimuth range of $[-62.4^\circ, +62.4^\circ]$ and elevation range of $[-48.5^\circ, +48.5^\circ]$, thus sufficiently large to activate the entire V1. We then computed texture-scramble d' values ($n = 4$ mice, 2 sessions each) finding higher d' values for the upper visual field. This new result is shown in Extended Data Fig. 3, described in the Results section (Ln 174) and in the Discussion (Ln 388) with a new Methods section as well "Texture-scramble discriminability gradient in V1".

2. I found the results in Fig 4c-f to be interesting but somewhat unconvincing. The analysis ultimately comes down to a correlation measured across four points without any real quantification. Is there a way to improve the statistical rigor of this analysis? For example, would it be possible to perform the same analysis with single texture images instead of texture families to allow more thorough quantification?

The previous Fig. 4c-f were indeed unclear about the underlying statistics, probably confusing the reviewer about the underlying analysis (we did not compute a correlation across four points), apologies for the confusion. We have significantly modified this figure and related text for improved clarity. We now show separate comparisons (boxplots with statistics) for neural d' , behavioral d' , and PS inter-cluster distances (new Fig. 4a-c), with the underlying distributions reflecting variability across 10 and 16 animals (for neural and behavioral d' , respectively), and 5th – 95th CIs for PS inter-cluster distances after bootstrapping and correcting for multiple comparisons (Fig. 4d.e). We are reporting detailed

statistical information (ANOVA and post-hoc Tukey HSD analysis) in the graphics and captions. Furthermore, we have removed the previous panel 4f since it was not adding new pieces of information relative to previous panels.

3. It would be nice to have an additional section in the Discussion of how the results in this manuscript compare to previous studies of texture/shape responses in lateral visual cortex in rodents from Zoccolan, Tolias, and Smith labs (references 55-58).

We have made a few edits in a related Discussion paragraph at Ln. 401 - 414. However, this section is already quite long, in which we discuss our results in the context of studies by Zoccolan, Tolias, and Smith labs. Given the limited space, further expanding this section would imply shrinking other paragraphs, making, in our opinion, the Discussion quite unbalanced relative to the topics addressed. However, if the reviewer judges that specific considerations are critically missing, we will attempt a further rewriting.

Minor typos/corrections:

1. The abstract has two difficult-to-parse sentences (lines 11-18) that should be rewritten for clarity.

We were not quite sure which specific phrasing was found to be difficult to parse. Therefore, we have rephrased the entire 2nd half of the abstract for improved clarity.

2. The sample size reported in the text (line 109) does not match the sample size reported in the figure (Fig 1g).

Thanks for spotting this inconsistency, we have corrected the mistake (n = 19).

3. Statistical comparisons in lines 115-116 are not clear.

We have added details to explain to which comparisons d' values refer to and moved this long sequence of statistical quantifications to the caption of Fig. 1f.

4. Lines 140, 296: I believe "neural activations" should be "neural activity"

Thanks, we have corrected both statements.

5. The blue/green colors For V1/LM in Fig. 3 are quite difficult to distinguish (at least on the printed page). I would recommend changing one of the colors to a more differentiable hue.

We are now using light and dark brown to also eliminate a previous inconsistency with the use of blue and red colors for scrambles and textures, of which we have modified the shades.

6. It is too late to change at this point, but the sample is dominated by males (~70% male). a more equal distribution would be preferable. Exact age range and mean (or median) age should be reported.

We have included this information in the Methods, section "Subjects".

7. Typo on the viral titer (line 783). Also, the virus injection volume is very large (~20x typical injection volume). The authors should consider multiple smaller injections or use of transgenic GCaMP mice for widefield imaging.

We have corrected the typo for the exponent. We find that slow injections of a large volume allow for a uniform labeling of dorsal visual areas. In our experience, this procedure ensures a baseline fluorescence and signal/noise that is reliably better than those of Tg lines we have tested over the years.

8. Lines 1006-1008: Typo at end of sentence.

9. Line 1015: Add reference number instead of in line reference.

10. Line 1021: Typo between sentences.

Thanks, points 8-10 have been corrected.

11. Extended Fig 2: Panel d is not described in caption.

We have corrected this oversight.

12. The statistical reporting is sometimes missing details. For each statistical test, the authors should describe the comparisons groups and report the statistical test, sample size, relevant statistic, exact p-value for all comparisons (often, one or more of these are missing). Similarly, box plots should be described in caption (can be described at end of figure if the same plot type is used throughout, but there is often no description to be found).

We have extensively revised the text to ensure statistical test, sample size, relevant statistic, exact p-value for all comparisons are always reported.

Overall, most of these points are fairly minor. This is a very nice paper and will make a nice contribution to the field.

Thanks again for the constructive comments and the appreciation of our work.

Reviewer #2

This study reveals that mice can discriminate families of visual textures, and provides insights into how this ability may derive from activity in two areas in their visual cortex (V1 and LM).

The main findings are that

(1) Mice discriminate among textures and between textures and statistically simpler matched stimuli.

(Fig 1)

(2) Neurons in V1 and especially LM respond on average more strongly to textures than to the matched stimuli (Fig 2, Fig 3)

(3) V1 and especially LM encode texture families in subspaces that are further apart when whose families differ more in statistics and are more discriminable perceptually (Fig 4).

These claims seem well supported by the data and the analyses, and the overall work is likely to be of interest to a wide range of scientists interested in high-level vision and mouse vision.

We appreciate the reviewer's recognition of our efforts.

MAJOR

The main suggestion is to explain more clearly what is different across textures and between textures and the scramble controls. A lot of the relevant information is in ED Fig 5. It would be useful to move it to the main text and to use graphics that better segregate the scrambles from the textures. Currently, the figure uses filled circles vs. filled stars, and these don't segregate well. Consider using open vs closed circles or some other notation where the two segregate.

We have moved the previous Extended Data Fig. 5 to the Results section, now Fig. 3g-j. We have also improved the graphics, as suggested. The Results section has been updated accordingly. In reference to a related comment by another referee, we have also introduced an analysis aiming to highlight the features in the images that are modulated by the energy components. We generated texture exemplars for all families imposing that all PS statistics other than the energy ones were fixed, matching those of a randomly chosen family, which was "scales" in the new Extended Data Fig. 7. By doing so, the generated images differed almost exclusively in the energy components. The outcome was akin to a "morphing" procedure, in which, perceptually, the various textures blended into the characteristic patterns of "scales". This agrees with psychophysical work showing that the energy components are the most relevant at predicting perceptual sensitivity in humans (e.g., Freeman et al., Nat. Neurosci., 2013). Together, these considerations have been included in the Results (Ln 228-231), Discussion (Ln 375-387) and Methods, at the end of the section "Texture synthesis".

In Fig 4 the paper devotes a lot of attention to the "energy" statistics, but it is not clear that those are so crucial. It is true that those statistics are very different across texture families and between texture vs. scramble (ED Fig 5a). However, this is also true of histogram skewness and kurtosis, the "marginal" statistics (ED Fig 5d). These are much simpler, as they could be computed directly from retinal images. Skewness, in particular, is simply the extreme value of ON and OFF channels. The latter is also called blackshot, and is a powerful discriminant of textures (Chubb et al, 1994, and <https://doi.org/10.1016/j.visres.2004.07.019>, and <https://doi.org/10.1167/3.9.60>).

It would be useful to know if skewness alone could be the cue that distinguishes textures from each other and from the scrambles. This is currently not clear from ED Fig 5d because that figure does not directly show the skewness and the kurtosis. Does that figure need to use PC1 and PC2? It would be so much clearer if it simply plotted skewness vs kurtosis. This would clarify whether the textures can be discriminated based on something as simple as blackshot.

We have addressed these comments with a few additional analyses. First, in reference to the comment about skewness and kurtosis, we have introduced a new figure panel, Fig. 3j, where we show a scatter plot with these two statistics. The plot is quite similar to the previous one with marginal statistics. As pointed out by the reviewer, some degree of clustering between texture families and textures-vs-scrambles can be observed in those representations. To examine whether mice relied on these differences, we generated a new set of images in which skewness and kurtosis were "jumbled up" (new Extended Data Fig. 3a, Methods, new section "Images with similar skewness and kurtosis"), thus with these variables being uninformative of the texture-family identity and texture-scramble category.

We then used these manipulated images in new behavioral experiments for texture-scramble and texture-texture discrimination tasks. The expectation was that if mice relied on these moments of the luminance-histogram distribution, removing this information (i.e., mixing-up skewness and kurtosis) would result in a lower performance. Instead, we observed that with the new set images, performance levels were comparable to those with the original images (new Extended Data Fig. 3b). We interpret this result as supporting evidence that mice are not relying on these statistics when performing the task. These considerations have been included in a new Results paragraph, Ln 135-143. This result seems consistent with previous considerations on the blackshot mechanism by Chubb et al., (Vis. Res. 2004), as cited by reviewer, in which the authors speculated that “...the primary function of this pre-attentive system has nothing to do with texture discrimination per se. Rather we suggest that the purpose of the blackshot system is to enable vision to be useful in shaded areas in an otherwise brightly illuminated field of view.”

MINOR

In the analysis shown in Fig 1f would be useful to connect the data for the 16 mice that were tested with all families, and use ANOVA or similar to test the hypothesis that some mice are better than others (regardless of texture) and some texture families are easier than others (regardless of mouse). Presumably it would show that the Rocks family was the hardest one. It would also be interesting to know if there is a significant interaction mouse/family.

In new analyses, we considered all individual training sessions for each mouse. The rocks family was consistently the most difficult to discriminate (repeated-measures ANOVA with Tukey HSD, significant for all pairwise comparisons). An analysis of the interactions did not reveal a clear significant trend, and only one mouse was particularly good, on average, but with a fair degree of variability. We have added connecting lines and detailed ANOVA statistics in the caption of Fig. 1f.

Reviewer #3

The authors present a study examining the processing of textures in the mouse visual system. The study first uses behavioral data showing to show that mice can distinguish textures from phase-scrambled control stimuli as well as between different families of textures. They then use both wide-field calcium imaging and two-photon calcium imaging to show that area LM contains neurons that are particularly sensitive to textures and that LM contains a population code that can distinguish between textures better than V1. They claim that LM represents textures in a more compact population space than V1. In general I think this is a well-executed study which is well-described and has a clear outcome, at least at the behavioral and single-cell levels. The results are a first step to showing the sensitivity of the mouse visual system to texture-stimuli and could in principle open the way to further studies examining the finer details of texture representation in LM and how learning affects these representations. I do have some caveats about the use of phase-scrambled controls and the strength of the claims about a more compact population code, aside from these I think this is a nice study.

Thanks for the appreciation of our work.

1) A major issue with using phase-scrambling to generate control images is that destroys all structure in the image, i.e. it removes all elongated contours from the image. It is therefore hard to determine what the mice were picking upon when discriminating between textures and scrambles – a mouse that has learnt to move the wheel when any elongated contour is present in the image would solve the task. Just visually examining the textures in Figure 1b would seem to confirm my worry as the ‘rocks’ texture has the least amount of high-contrast elongated contours compared to the other texture families and is also the hardest to discriminate. The fact that the animals could also discriminate between texture families is more convincing and suggests that the animals are picking up on texture cues rather than simple structures. But still, I would like to see some acknowledgement and discussion of this issue by the authors.

We largely agree with the reviewer’s observation, with some caveats. As shown in Extended Data Fig. 1c, the average orientation power of textures and scrambles is equalized, indicating that, within the examined spatial-frequency range (of perceptual relevance to the mice, up to 0.8–1.0 cpd) “overall” oriented cues cannot be used to solve the task. Furthermore, texture rotations have been introduced as well. However, the reviewer is correct when noting that elongated “wiggly” structures and patterns are overall removed by the phase scrambling procedure. Nonetheless, these “structures and patterns” are repeated features in an image that largely shape the high-order spatial correlations and are prominently texture-defining. Therefore, it is the ability of mice to detect and discriminate these elongated and wiggly features (possibly among other features) that is taken as a proxy for texture detection and discrimination. We have reported these observations in a new Discussion paragraph, Ln 381-390.

2) Related to this. It would be useful for a reader to get an intuition about what is being captured by the PS statistics. I would like to see example images which vary along one dimension of this statistical space while keeping the others constant. Perhaps the ‘contour’ dimension I refer to above is what is being captured by the energy dimension? It would in any case be very useful to gain an intuition in how changes along the energy dimension alter the textures.

It is rather challenging to thoroughly address this point, and we are not aware of a computationally tractable approach to tackle this point using CNNs as generative models. However, we attempted some new analysis using the PS generative model, aiming to highlight image features that are modulated by the energy components. To this end, we generated texture exemplars for each family imposing that all PS statistics other than the energy ones were fixed, constrained to be those of a randomly chosen family— “scales” in the example of Extended Data Fig. 7 (Methods, new paragraph at the end of the section “Texture synthesis”). By doing so, the generated images differed (almost) exclusively in the energy components. The perceptual outcome was quite difficult to interpret, akin to a “partial morphing” of features, with some aspects of the images perceptually blending into the characteristic patterns of “scales”. This agrees with psychophysical work showing that in humans the energy components are the most relevant at predicting perceptual sensitivity of texture images (e.g., Freeman et al., Nat. Neurosci., 2013). These considerations have been added to the Results (Ln 228-231). As an additional note, in this Discussion we are referring to human psychophysical judgments, which have been explicitly incorporated in models of texture synthesis (e.g., eq. 14 in Ding et., al. 2020). However, it remains challenging to relate

these judgments to mouse visual perception, when trying to infer heuristics adopted by mice in these tasks (Discussion, Ln 381-390).

3) I'm sure what to make of Figures 4j-l. The authors report a significant difference in cluster radii and inter-cluster distance between LM and V1 in the text. But in Figure 4l it states that the percentage-wise change is not significant. Given that the 'compactness' of the population representation in LM is one of the findings highlighted in the abstract this seems not to be a very robust effect. The scatter-plots also don't seem very convincing. I would recommend using less strong language concerning these population effects and adding some more nuance to the abstract and discussion of these results.

Apologies for the unclear explanations: the y-axis in former Fig. 4l indicated the "effect size" (expressed as a % change) for differences in radii and distances between V1 and LM. The effect size for the radii was found to be statistically indistinguishable from that of distances, but both effects were significantly different from zero. We acknowledge that the presentation of these results was confusing, hence we have modified several panels in this figure and added a few new statistical analyses, as detailed in response to the following comment.

Also – the graphs in Figure 4j-k show multiple points per mouse, how was the paired t-test statistic calculated? With an n of 10 (mice, as reported in the text, did they then average across families?) or an n of 40 (mice x texture family). This is really grouped data with multiple points per mouse and so a repeated measures ANOVA or linear mixed effects model would be more appropriate.

The paired t-test is computed with an $n = 40$, now corrected in the text. As suggested, we have computed new statistics using repeated-measures ANOVA. This test is based again on $n = 40$ measurements (10 mice x 4 texture families), using as within-subject variables the cortical areas (V1, LM) and four texture families. Main effects for radii were found to be significant both for areas and families, with a significant interaction as well, indicating an effect of area on the representations of the different families. Instead, for the distances, the area and family effects were not significant, with only the interaction being significant. Overall, these results support the finding of a decrease in radii, with a decrease in the inter-cluster distances being significant according to a paired t-test but not with repeated-measures ANOVA. Therefore, we have rewritten related sentences in the Abstract, Results, and Discussion, emphasizing the decrease in radii and significantly downplaying the effect on distances.

As the authors point out, the fact that the radii and inter-cluster distance both decrease could have had counter-acting effects on the performance of the decoder, but this doesn't appear to be the case.

As clarified above, the effect on inter-cluster distances has been significantly downplayed. The significant interaction term in the R-ANOVA test demonstrates that distances between families are not independent of area, with the paired t-test indicating that this interaction involves a decrease in inter-cluster distances in LM relative to V1. However, the most statistically robust effect is the one on the reduction in radii, as reflected in the corrected narrative.

These two measures could be directly combined into a measure such as Mahalanobis distance or a multi-dimensional d-prime which would directly measure the distance between clusters in units of

their standard deviation (it is not 100% clear to me if the inter-cluster distance has been normalized by the radii of the clusters, on line 301 they state that it is the Euclidean distance, but in lines 1179-1184 they talk about dividing by the mean radii, which is correct?). Some clarification is required here.

The accuracy of the multinomial logistic classifier is indeed a combined measure of radii and distances, and also of the “overall shape” of the distributions in the PCA representational space. Subsequently, we wanted to identify features in these representations that most significantly contributed to the improved accuracy of the classifier. To this end, we performed separate analyses for radii and distances. However, since we used logistic regression instead of, for example, LDA (which could have been directly related to a multidimensional d-prime measure), we agree that reporting a combined measure other than the classifier’s performance can provide complementary information. Hence, as suggested, we computed Mahalanobis-distance values, and found a significant increase in distance in LM compared to V1, in agreement with the classifier performance (Ln 303-307). Finally, apologies for the confused explanations in the Methods which we have rewritten for clarity (Ln 1315-1323).

4) Were the mice water-restricted? Perhaps I missed it, but this procedure should be described in the methods.

Yes, they were. This information is now reported in the Methods, Ln 868.

Minor points

What is the function of the contours in Figure 4j-k? The scatter-plots would be easier to interpret without them.

We introduced this KDE representation because some dots were overlapping; we have now removed it as suggested and increased the size of the dots as well (new Fig. 4i).

Line 29: Should be ‘Synthesized’

Thanks, fixed.

Line 115: ‘d’ = 1.4 ± 0.14, n = 16, d’ > 1, p = 0.016; p = 3 x 10⁻⁵, n = 15’ It isn’t clear what the different p-values and n-values refer to.

We have moved this information to the caption, and concisely added clarifying details.

Line 721: ‘One-ay’ typo.

Thanks, fixed.

Line 734: ‘The behavioral discriminability as in (a)’ Should be panel (b). Also what re inter-cluster distances?

Thanks, we have corrected the panel’s reference. For the inter-cluster distances, we have added a new panel (c).

Line 819: ‘Their self-head was fixed’ – should be ‘their head was self-fixed’.

Thanks, fixed.

Line 1008: ‘retinotopic’ missing word.6

Thanks, fixed.

REVIEWERS' COMMENTS

Reviewer #1 (Remarks to the Author):

The authors have addressed all of my concerns and I recommend the manuscript for publication once the other reviewers are satisfied. This is a nicely done study, and I look forward to seeing it in print.

Reviewer #2 (Remarks to the Author):

This excellent paper has benefited from the revisions and this this reviewer has no substantial further comments.

Some additional minor suggestions are below in case they are useful:

-The beginning of the abstract is misleading, in the sense that it leads the reader to expect the wrong thing. The first sentence frames the paper as dealing with “communication between neurons”, which it does not. The second sentence mentions a “circuit-level understanding”, which the paper does not deliver. It would be best to rewrite those sentences to frame the paper in the correct direction. Indeed, this paper does deliver a lot of results, but it is not about communication between neurons nor about the underlying circuits. Consider taking sentences from your own Introduction, which is very good.

-Line 188: “Number of cells responding to textures is higher in LM”. Shouldn’t this say “Fraction” instead of “Number”?

-Line 217. This section title is in a different style (the previous one was declarative, this one isn’t). Would be good to harmonize the titles.

-Fig 3d-f it would be great if these could be rearranged in a matrix with 4 rows (one per texture type) and 3 columns (d' for V1, d' for LM, and comparison between the two). Crucially, the x axes for the first two columns should be the same. For instance, if the x axis for Scales goes from -1 to 3 for LM, it should cover the same range also for V1. This would help the reader compare the results for V1 and LM.

Reviewer #3 (Remarks to the Author):

The authors have addressed my main concerns and I'm happy with the revision, I just have a couple of minor points that could be addressed in the final manuscript.

- It's confusing to report both the paired t-test and ANOVA results in Figure 4j. The paired t-test isn't a great statistic here as it ignores the fact that the data is grouped by mouse and is therefore too liberal. It would be better to separate out the two bars in Figure 4j into four bars per area to allow the reader to judge the interaction effect and simply report the results of the rep. measures ANOVA.

- Please provide full details of the water restriction in the methods e.g. the typical volume of fluid obtained in training sessions, the minimum amount allowed per day, in the weekend, welfare monitoring.

Reviewer #2 (Remarks to the Author):

This excellent paper has benefited from the revisions and this this reviewer has no substantial further comments.

We thank the reviewer for the constructive comments which have helped improve our study.

Some additional minor suggestions are below in case they are useful:

-The beginning of the abstract is misleading, in the sense that it leads the reader to expect the wrong thing. The first sentence frames the paper as dealing with “communication between neurons”, which it does not. The second sentence mentions a “circuit-level understanding”, which the paper does not deliver. It would be best to rewrite those sentences to frame the paper in the correct direction. Indeed, this paper does deliver a lot of results, but it is not about communication between neurons nor about the underlying circuits. Consider taking sentences from your own Introduction, which is very good.

We agree with the reviewer that our study is not directly addressing questions about the communication between neurons, hence we now use the more general term ‘activity’. However, regarding the comment on the circuit-level understanding, we believe that our results can indeed add to this descriptive level. In particular, the result of the neuronal populations in LM being associated to more compact representations of texture stimuli, can, in our view, be described as a result at the level of neuronal circuits, emerging from the collective activity of ensembles of neurons in V1 and LM – as opposed, for example, to an overall area effect revealed by an fMRI study. Hence, we’d rather keep this phrasing.

-Line 188: “Number of cells responding to textures is higher in LM”. Shouldn’t this say “Fraction” instead of “Number”?

Thanks, we have corrected this imprecision, now ‘Proportion of cells responding to textures in V1 and LM’.

-Line 217. This section title is in a different style (the previous one was declarative, this one isn’t). Would be good to harmonize the titles.

Section titles are declarative except for the one about the proportion of cells, which was descriptive as noted by the reviewer. We have changed it to declarative to have all titles in the same style.

-Fig 3d-f it would be great if these could be rearranged in a matrix with 4 rows (one per texture type) and 3 columns (d’ for V1, d’ for LM, and comparison between the two). Crucially, the x axes for the first two columns should be the same. For instance, if the x axis for Scales goes from -1 to 3 for LM, it should cover the same range also for V1. This would help the reader compare the results for V1 and LM.

We have implemented these changes, although the x-range is slightly different between V1 and LM – but consistently so across families – without the by-eye comparison between areas suffering from this difference, in our opinion. Panel (f) delivers the key statistical quantification, with all data plotted using a consistent y-range.

Reviewer #3 (Remarks to the Author):

The authors have addressed my main concerns and I'm happy with the revision, I just have a couple of minor points that could be addressed in the final manuscript.

- It's confusing to report both the paired t-test and ANOVA results in Figure 4j. The paired t-test isn't a great statistic here as it ignores the fact that the data is grouped by mouse and is therefore too liberal. It would be better to separate out the two bars in Figure 4j into four bars per area to allow the reader to judge the interaction effect and simply report the results of the rep. measures ANOVA.

We have implemented the suggested changes. For the graphics, we have adopted a slightly different format; however, by keeping the y-limits consistent across panels the suggested comparison can easily be visualized.

- Please provide full details of the water restriction in the methods e.g. the typical volume of fluid obtained in training sessions, the minimum amount allowed per day, in the weekend, welfare monitoring.

We have added these details at the end of the Methods section "Behavioral training procedure".